# Privacy Amplification via Compression: Achieving the Optimal Privacy-Accuracy-Communication Trade-offs in Distributed Mean Estimation

**Wei-Ning Chen**
Stanford University
`wnchen@stanford.edu`

**Dan Song**
Stanford University
`songdan@stanford.edu`

**Ayfer Özgür**
Stanford University
`aozgur@stanford.edu`

**Peter Kairouz**
Google Research
`kairouz@google.com`

## Abstract

Privacy and communication constraints are two major bottlenecks in federated learning (FL) and analytics (FA). We study the optimal accuracy of mean and frequency estimation (canonical models for FL and FA respectively) under joint communication and $(\varepsilon, \delta)$-differential privacy (DP) constraints. We consider both the central and the multi-message shuffled DP models. We show that in order to achieve the optimal $\ell_2$ error under $(\varepsilon, \delta)$-DP, it is sufficient for each client to send $\Theta\left(n \min\left(\varepsilon, \varepsilon^2\right)\right)$ bits for FL and $\Theta\left(\log\left(n \min\left(\varepsilon, \varepsilon^2\right)\right)\right)$ bits for FA to the server, where $n$ is the number of participating clients. Without compression, each client needs $O(d)$ bits and $O(\log d)$ bits for the mean and frequency estimation problems respectively (where $d$ corresponds to the number of trainable parameters in FL or the domain size in FA), meaning that we can get significant savings in the regime $n \min\left(\varepsilon, \varepsilon^2\right) = o(d)$, which is often the relevant regime in practice.

We propose two different ways to leverage compression for privacy amplification and achieve the optimal privacy-communication-accuracy trade-offs. In both cases, each client communicates only partial information about its sample and we show that privacy is amplified by randomly selecting the part contributed by each client. In the first method, the random selection is revealed to the server, which results in a central DP guarantee with optimal privacy-communication-accuracy trade-offs. In the second method, the random data parts from the clients are shuffled by a secure shuffler resulting in a multi-message shuffling scheme with the same optimal trade-offs. As a result, we establish the optimal three-way trade-offs between privacy, communication, and accuracy for both the central DP and multi-message shuffling frameworks.

## 1 Introduction

In the basic setting of federated learning (FL) [68, 64, 61] and analytics (FA), a server wants to execute a specific learning or analytics task on raw data that is kept on clients' devices. Consider, for example, model updates in FL or histogram estimation in FA, both of which can be modeled as a distributed mean estimation problem. Clients communicate targeted messages to the server and the privacy of the users' data is ensured (in terms of explicit differential privacy (DP) [38] guarantees) by adding carefully calibrated noise to the computed mean at the server before releasing it to downstream modules (e.g., the server computes the average model update and corrupts it with the addition of

37th Conference on Neural Information Processing Systems (NeurIPS 2023).

noise). This is called the trusted server or central DP model, as it entrusts the central server with privatization and is one of the most common ways in which federated learning and analytics are implemented today [1].

In this paper, we ask the following question: given that the server needs to privatize the mean, can the clients communicate "less information" to the server? More precisely, can we leverage the fact that the server only needs to output a noisy (approximate) estimate of the mean to reduce the communication load without sacrificing accuracy? In recent years, there has been significant interest in the central DP model [1] as well as communication efficiency and privacy for FL and FA under different models, including local DP [78, 63, 58, 80, 17, 6, 16, 32], shuffle [40, 43] and distributed DP [9, 59, 8, 33, 34]; however, this basic question has remained open.

One natural way to reduce communication is to have clients communicate only partial information about their samples. For example, in the case of model updates, each client can update only a subset of the model coefficients. In histogram estimation, information about a client's sample can be "split" into multiple parts, and the client can communicate only one part. However, this results in less information at the server, or effectively fewer samples to estimate the target quantity, e.g., each model coefficient is now updated only by a subset of the clients. A quick calculation reveals that this increases the sensitivity of the estimate to each user's sample and therefore requires the addition of larger noise at the server to achieve the same privacy level. Hence, reducing communication reduces accuracy for the same privacy guarantee.

We circumvent this challenge with a simple but insightful observation: when each client communicates only partial information about its sample, we can amplify privacy by randomly selecting the part contributed by each client. This random selection is hidden from a downstream module which has only access to the estimate revealed by the server, which leads to privacy amplification. Privacy amplification by subsampling has been studied in [66, 14] but usually refers to the selection of a random subset of the clients (from a larger pool of available clients). In our case, it is the "piece of information" that is randomly selected at each client. We call this gain privacy amplification via compression as it emerges from the fact that each client does not fully communicate its sample. We use it to establish the optimal communication-privacy-accuracy trade-off for central DP. Note that this same type of gain cannot be leveraged in the local DP model where the server is untrusted. The server needs to know the random selection at each client to construct an estimate. Indeed, in the local DP model, the privacy-accuracy trade-off is known to be significantly worse than the central DP model (see Table 1).

This naturally leads to a follow-up question: can we leverage privacy amplification via compression and achieve the same three-way trade-off by using secure aggregation [33] and shuffling [40] type models which hide information from the server? For secure aggregation, the three-way trade-off has been studied in [31] and the communication cost is significantly larger than the communication cost for central DP proved in this paper (see Table 1). For shuffling, the optimal communication cost has been posed as an open problem in [31]. We resolve this problem by showing that the optimal central DP trade-off can also be achieved with a multi-message shuffling scheme, establishing the optimal communication cost. (We note that a similar result has been concurrently and independently proved in [54] under the shuffle DP setting.) As before, our scheme leverages a privacy amplification gain. Each client communicates partial information about its sample; the identity of the message is erased by the secure shuffler, and hence the untrusted server does not know which part is contributed by each client. We show that to achieve the optimal trade-off, it is critical for each client to split its information into multiple messages and employ multiple shuffling rounds by carefully splitting the privacy budget across different rounds. In contrast, the linearity of secure aggregation requires all participating clients to communicate consistent information (same parts), hence precluding privacy amplification by compression. See Table 1 for a detailed comparison.

**Our contributions.** We study distributed mean and frequency estimation as canonical building blocks for FL and FA. We consider both the central DP and the multi-message shuffling models. We characterize the order-optimal privacy-accuracy-communication trade-offs for distributed mean estimation and provide an achievable scheme for frequency estimation (in Appendix A) under the central DP model. Our results reveal that privacy and communication efficiency can be achieved simultaneously with no additional penalty for accuracy. In particular, we show that $\tilde{O}\left(n\min\left(\varepsilon, \varepsilon^2\right)\right)$

---

[1]We assume a trusted service provider who applies the DP mechanism faithfully. This can be enforced by implementing the DP mechanism inside of a remotely attestable trusted execution environment [11].

| | Communication (bits) | $\ell_2^2$ error |
|---|---|---|
| Local DP [32, 42] | $\Theta\left(\lceil\varepsilon\rceil\right)$ | $\Theta\left(\frac{d}{n\min(\varepsilon^2,\varepsilon)}\right)$ |
| Distributed DP (with SecAgg) [33] | $\tilde{O}\left(n^2\min\left(\varepsilon,\varepsilon^2\right)\right)$ | $\Theta\left(\frac{d}{n^2\min(\varepsilon^2,\varepsilon)}\right)$ |
| Central DP (Theorem 4.4) | $\tilde{O}\left(n\min\left(\varepsilon,\varepsilon^2\right)\right)$ | $O\left(\frac{d\log d}{n^2\min(\varepsilon^2,\varepsilon)}\right)$ |
| Shuffle DP (Theorem 5.3, [54]) | $\tilde{O}\left(n\log(d)\min\left(\varepsilon,\varepsilon^2\right)\right)$ | $O\left(\frac{d}{n^2\min(\varepsilon^2,\varepsilon)}\right)$ |

Table 1: Comparison of the communication costs of $\ell_2$ mean estimation under local, distributed, central, and shuffle DP (with $\delta$ terms hidden). Compared to local DP, we see that error under central DP decays much faster (e.g., $1/n^2$ as opposed to $1/n$); compared to distributed DP with secure aggregation, our schemes achieve similar accuracy but saves the communication cost by a factor of $n$.

and $\tilde{O}\left(\log\left(n\min\left(\varepsilon,\varepsilon^2\right)\right)\right)$ bits of (per-client) communication are sufficient to achieve the order-optimal error under $(\varepsilon,\delta)$-privacy for mean and frequency estimation respectively, where $n$ is the number of participating clients. Without compression, each client needs $O(d)$ bits and $\log d$ bits for the mean and frequency estimation problems respectively (where $d$ is the number of trainable parameters in FL or the domain size in FA), which means that we can get significant savings in the regime $n\varepsilon^2 = o(d)$ (assuming $\varepsilon = O(1)$). We note that this is often the relevant regime not only for cross-silo but also for cross-device FL/FA. For instance, in practical FL, $d$ usually ranges from $10^6$ to $10^9$, and $n$, the *per-epoch* sample size, is usually much smaller (e.g., of the order of $10^3$ to $10^5$). For distributed mean estimation, we show that the central DP trade-off can also be achieved with a multi-message shuffling scheme (within a $\log d$ factor in communication cost). Hence our paper establishes the three-way trade-off between privacy, communication, and accuracy for both the central DP and multi-message shuffling frameworks, both of which were open problems in the prior literature. Compared with local DP where 1 bit is sufficient when $\varepsilon = O(1)$, this shows that central/shuffling DP has a larger communication cost but can achieve much smaller error (by a factor of $n$) and hence is usually preferable in practical applications. Compared with distributed DP where the server aggregates local (encoded) messages with secure multi-party computation (e.g., [23, 8, 34]), we can improve the communication cost by a factor of $n$, therefore showing that the communication cost can be reduced with a trusted server or shuffler. We summarize the comparisons of our main results to local and distributed DP in Table 1.

**Notation.** Throughout this paper, we use $[m]$ to denote the set of $\{1,...,m\}$ for any $m \in \mathbb{N}$. Random variables (vectors) $(X_1,...,X_m)$ are denoted as $X^m$. We also make use of Bachmann-Landau asymptotic notation, i.e., $O, o, \Omega, \omega,$ and $\Theta$.

## 2 Problem Formulation

We first present the distributed mean estimation (DME) [74] problem under differential privacy. Note that DME is closely related to federated learning with SGD (or similar stochastic optimization methods, such as FedAvg [68]), where in each iteration, the server updates the global model by a noisy mean of the local model updates. This noisy estimate is typically obtained by using a DME scheme, and thus one can easily build a distributed DP-SGD scheme (and hence a private FL scheme) from a differentially private DME scheme. Moreover, as shown in [49], as long as we have an unbiased estimate of the gradient at each round, the convergence rates of SGD (or DP-SGD) depend on the $\ell_2$ estimation error.

**Distributed mean estimation.** Consider $n$ clients each with local data $x_i \in \mathbb{R}^d$ that satisfies $\|x_i\|_2 \le C$ for some constant $C > 0$ (one can think of $x_i$ as a clipped local gradient). A server wants to learn an estimate $\hat{\mu}$ of the mean $\mu(x^n) \triangleq \frac{1}{n}\sum_i x_i$ from $x^n = (x_1,\ldots,x_n)$ after communicating with the $n$ clients. Toward this end, each client locally compresses $x_i$ into a $b$-bit message $Y_i = \text{enc}_i(x_i) \in \mathcal{Y}$ through a local encoder $\text{enc}_i : \mathcal{X} \mapsto \mathcal{Y}$ (where $|\mathcal{Y}| \le 2^b$ and sends it to the central server, which upon receiving $Y^n = (Y_1,\ldots,Y_n)$ computes an estimate $\hat{\mu} = \text{dec}(Y^n)$ that satisfies the following differential privacy:

**Definition 2.1** (Differential Privacy)**.** *The mechanism $\hat{\mu}$ is $(\varepsilon, \delta)$-differentially private if for any neighboring datasets $x^n := (x_1, ..., x_i, ..., x_n)$, $x'^n := (x_1, ..., x'_i, ..., x_n)$, and measurable $\mathcal{S} \subseteq \mathcal{Y}$,*

$$\Pr\{\hat{\mu} \in \mathcal{S} | x^n\} \leq e^\varepsilon \cdot \Pr\{\hat{\mu} \in \mathcal{S} | x'^n\} + \delta,$$

*where the probability is taken over the randomness of $\hat{\mu}$.*

Our goal is to design schemes that minimize the $\ell_2^2$ estimation error:

$$\min_{(\mathsf{enc}_1(\cdot), ..., \mathsf{enc}_n(\cdot), \mathsf{dec}(\cdot))} \max_{x^n} \mathbb{E}\left[\|\hat{\mu}\left(\mathsf{enc}_1(x_1), ..., \mathsf{enc}_n(x_n)\right) - \mu(x^n)\|_2^2\right],$$

subject to $b$-bit communication and $(\varepsilon, \delta)$-DP constraints.

**Distributed frequency estimation.** Similarly, frequency estimation can also be formulated as a mean estimation problem but with sparse (one-hot) vectors. Let each user $i$ hold an item $x_i$ in a size $d$ domain $\mathcal{X}$. The server aims to estimate the histogram of the $n$ items. Without loss of generality, we can assume that $\mathcal{X} := \{e_1, ..., e_d\} \in \{0, 1\}^d$ (where $e_j$ is the $j$-th standard basis vector in $\mathbb{R}^d$), i.e., each item is expressed as a one-hot vector. Then, the histogram of the $n$ items can be expressed as $\pi(x^n) := \sum_{i \in [n]} x_i$. Similar to the mean estimation problem, clients locally compute and then send $Y_i = \mathsf{enc}_i(x_i) \in \mathcal{Y}$ (for some $\mathcal{Y}$ such that $|\mathcal{Y}| \leq 2^b$), and the central server computes the estimate $\hat{\pi} = \mathsf{dec}(y^n)$. Our goal is to design schemes that minimize the $\ell_2^2$ or $\ell_1$ error[2]:

$$\min_{(\mathsf{enc}_1(\cdot), ..., \mathsf{enc}_n(\cdot), \mathsf{dec}(\cdot))} \max_{x^n} \mathbb{E}\left[\|\hat{\pi}\left(\mathsf{enc}_1(x_1), ..., \mathsf{enc}_n(x_n)\right) - \pi(x^n)\|\right],$$

subject to communication and DP constraints (where $\|\cdot\|$ can be $\ell_1$ or $\ell_2^2$).

## 3  Related Works

**Federated learning and distributed mean estimation.** Federated learning [64, 68, 60] emerges as a decentralized machine learning framework that provides data confidentiality by retaining clients' raw data on edge devices. In FL, communication between clients and the central server can quickly become a bottleneck [68], so previous works have focused on compressing local model updates via gradient quantization [68, 10, 48, 74, 79, 77, 24], sparsification [18, 56, 41]. To further enhance data security, FL is often combined with differential privacy [38, 1, 9]. Among these works, [**?** ] also employs gradient sparsification (or gradient subsampling) to reduce the problem dimensionality. However, the sparsification takes place *after* the aggregation of local gradients, so the randomness introduced during sparsification cannot be leveraged to amplify the differential privacy guarantee. As a result, this approach leads to a suboptimal trade-off between privacy and communication compared to our scheme.

Note that in this work, we consider FL (or more specifically, the distributed mean estimation) under a *central*-DP setting where the server is trusted, which is different from the local DP model [63, 37, 70, 76, 22, 32] and the distributed DP model with secure aggregation [23, 21, 59, 8, 33, 34].

A key step in our mean estimation scheme is pre-processing the local data via Kashin's representation [67]. While various compression schemes, based on quantization, sparsification, and dithering have been proposed in the recent literature, Kashin's representation has also been explored in a few works for communication efficiency [47, 73, 29, 71] and for LDP [42] and is particularly powerful in the case of joint communication and privacy constraints as it helps spread the information in a vector evenly in every dimension.

**Distributed frequency estimation and heavy hitters.** Distributed frequency estimation (a.k.a. histogram estimation) is another canonical task that has been heavily studied under a distributed setting with DP. Prior works either focus on 1) the local DP model with or without communication constraints, e.g., [20, 19, 25, 26, 57] (under an $\ell_\infty$ loss for heavy hitter estimation) and [58, 80, 76, 6, 5, 32, 46, 72, 45] (under an $\ell_1$ or $\ell_2$ loss), or 2) the central DP model *without* communication constraints [38, 52, 65, 27, 13, 81, 36]. As suggested in [37, 3, 2, 4, 16], compared to central DP, local DP models usually incur much larger estimation errors and can significantly decrease the utility. In this work, we consider central DP but with explicit communication constraints.

---

[2]Note that the $\ell_1$ error corresponds to the total variation distance between the true and estimated frequency vectors.

**Local DP with shuffling.** A recent line of works [40, 35, 12, 43, 50, 51] considers *shuffle*-DP, showing that one can significantly boost the central DP guarantees by randomly shuffling local (privatized) messages. In this work, we show that the same shuffling technique can be used to achieve the optimal central DP error with nearly optimal communication cost. Therefore, we can obtain the same level of central DP with small communication costs while weakening the security assumption: achieving the optimal communication cost (under central DP) only requires a secure shuffler (as opposed to a fully trusted central server).

## 4 Distributed Mean Estimation

In this section, we present a mean estimation scheme that achieves the optimal $\tilde{O}_\delta \left( \frac{C^2 d}{n^2 \varepsilon^2} \right)$ error under $(\varepsilon, \delta)$-DP while only using $\tilde{O}(n\varepsilon^2)$ bits of per-client communication.

We first consider a slightly simpler, discrete setting with $\ell_\infty$ geometry (as opposed to the $\ell_2$ mean estimation stated in Section 2): assume each client observes $x_i \in \{-c, c\}^d$ where $c > 0$ is a constant, and a central server aims to estimate the mean $\mu(x^n) := \frac{1}{n} \sum_{i=1}^n x_i$ by minimizing the $\ell_2^2$ error subject to the privacy and communication constraints. We argue later that solutions to the above $\ell_\infty$ problem can be used for $\ell_2$ mean estimation by applying Kashin's representation.

To solve the aforementioned $\ell_\infty$ mean estimation problem, first observe that each client's local data can be expressed in $d$ bits since each coordinate of $x_i$ can only take values in $\{c, -c\}$. To reduce the communication load to $o(d)$ bits, each client adopts the following subsampling strategy: for each coordinate $j \in [d]$, client $i$ chooses to send $x_i(j)$ to the server with probability $\gamma$. We assume that this subsampling step is performed with a seed shared by the client and the server[3], hence the server knows which coordinates are communicated by each client. Therefore upon receiving the client messages, it can compute the mean of each coordinate and privatize it by adding Gaussian noise. The key observation we leverage is that the randomness in the compression algorithm can be used to amplify privacy or equivalently reduce the magnitude of the Gaussian noise that is needed for privatization. Note that such randomness needs to be kept private from an adversary as the privacy guarantee of the scheme relies on it.

We summarize the scheme in Algorithm 1 and state its privacy and utility guarantees in the following theorem.

**Theorem 4.1** ($\ell_\infty$ mean estimation.)**.** *Let $x_1, ..., x_n \in \{-c, c\}^d$ and $\varepsilon, \delta > 0$. There exists a*

$$\sigma^2 = O\left( \frac{c^2 \log(1/\delta)}{n^2 \gamma^2} + \frac{c^2 d(\log(d/\delta) + \varepsilon) \log(d/\delta)}{n^2 \varepsilon^2} \right) \tag{1}$$

*such that Algorithm 1 is $(\varepsilon, \delta)$-DP and the $\ell_2^2$ estimation error of $\hat{\mu}$ is at most*

$$\mathbb{E}\left[ \|\hat{\mu} - \mu\|_2^2 \right] \leq \frac{dc^2}{n\gamma} + d\sigma^2$$

$$= O\left( \frac{d^2 c^2}{nb} + \frac{d^3 c^2 \log(d/\delta)}{n^2 b^2} \right. \tag{2}$$

$$\left. + \frac{c^2 d^2 (\log(1/\delta) + \varepsilon) \log(d/\delta)}{n^2 \varepsilon^2} \right). \tag{3}$$

*In addition, the (average) per-client communication cost is $\gamma d = b$ bits, and Algorithm 1 yields an unbiased estimator of $\mu$.*

---

**Algorithm 1** Coordinate Subsampled Gaussian Mechanism (CSGM)

---

**Input:** users' data $x_1, ..., x_n$, sampling parameters $\gamma := b/d$, DP parameters $(\varepsilon, \delta)$.
**Output:** mean estimator $\hat{\mu}$.
**for** user $i \in [n]$ **do**
    **for** coordinate $j \in [d]$ **do**
        Draw $Z_{i,j} \overset{\text{i.i.d.}}{\sim} \text{Bern}(\gamma)$.
        **if** $Z_{i,j} = 1$ **then**
            Send $x_i(j)$ to the server.
        **end if**
    **end for**
**end for**
**for** coordinate $j \in [d]$ **do**
    Server computes the average $\hat{\mu}_j := \frac{1}{n\gamma} \sum_{i:Z_{ij}=1} x_i(j) + N(0, \sigma^2)$, where $\sigma^2$ is computed according to (1) in Theorem 4.1.
**end for**
**Return:** $\hat{\mu} := (\hat{\mu}_1, \hat{\mu}_2, ..., \hat{\mu}_d)$.

---

**Remark 4.2** (Unbiasedness)**.** *Note that for mean estimation, we usually want the final mean estimator to be unbiased since standard convergence analyses of SGD [49] require an unbiased estimate of the true gradient in each optimization*

---

[3]In practice, such randomness can be agreed by both sides ahead of time, or it can be generated by the server and communicated to each client.

*round. Given that our proposed mean estimation schemes (Algorithm 1 and Algorithm 2 in the next section) are all unbiased, we can combine them with SGD/federated averaging and readily apply [49] to obtain a convergence guarantee for the resulting communication-efficient DP-SGD.*

For the $\ell_2$ mean estimation task formulated in Section 2, we pre-process local vectors by first computing their Kashin's representations and then performing randomized rounding [62, 75, 42, 32]. Specifically, if $x_i$ has $\ell_2$ norm bounded by $C$, then its Kashin's representation (with respect to a tight frame $K \in \mathbb{R}^{d \times D}$ where $D = \Theta(d)$) $\tilde{x}_i$ has bounded $\ell_\infty$ norm: $\|\tilde{x}_i\|_\infty \le c = O\left(\frac{C}{\sqrt{d}}\right)$ and satisfies $x_i = K\tilde{x}_i$. This allows us to convert the $\ell_2$ geometry to an $\ell_\infty$ geometry. Furthermore, by randomly rounding each coordinate of $\tilde{x}_i$ to $\{-c, c\}$ (see for example [32]), we can readily apply Algorithm 1 and obtain the following result for $\ell_2$ mean estimation as a corollary:

**Corollary 4.3** ($\ell_2$ mean estimation). *Let $x_1, ..., x_n \in \mathcal{B}_2(C)$ (i.e., $\|x_i\|_2 \le C$ for all $i \in [n]$). Then for any $\varepsilon, \delta > 0$, Algorithm 1 combined with Kashin's representation and randomized rounding yields an $(\varepsilon, \delta)$-DP unbiased estimator for $\mu$ with $\ell_2^2$ estimation error bounded by*

$$O\left(\underbrace{\frac{dC^2}{nb} + \frac{C^2 d^2 \log(1/\delta)}{n^2 b^2}}_{(\alpha)} + \underbrace{\frac{C^2 d(\log(d/\delta) + \varepsilon)\log(d/\delta)}{n^2 \varepsilon^2}}_{(\beta)}\right). \tag{4}$$

The first term $(\alpha)$ in the estimation error in Corollary 4.3 is the error due to compression, and the second term $(\beta)$ is the error due to privatization (which is order-optimal under $(\varepsilon, \delta)$-DP up to an additional $\log(d/\delta)$ factor as we discuss in Section 4.2). In particular, if we ignore the poly-logarithmic terms and assume $\varepsilon = O(1)$, the privatization error $(\beta)$ can be simplified to $\tilde{O}\left(\frac{dC^2}{n^2\varepsilon^2}\right)$, which dominates the total $\ell_2^2$ error when $b = \tilde{\Omega}_\delta\left(\max\left(n\varepsilon^2, \sqrt{d}\varepsilon\right)\right)$, i.e. in this regime the total $\ell_2^2$ error is order-wise equal to the optimal centralized DP error $(\beta)$. This implies that no more than $b = \tilde{\Omega}_\delta\left(\max\left(n\varepsilon^2, \sqrt{d}\varepsilon\right)\right)$ bits per client are needed to achieve the order-optimal $\ell_2^2$ error.

In the next section, we introduce a modification to Algorithm 1, which allows the removal of the $\Omega\left(\sqrt{d}\varepsilon\right)$ term in the communication cost.

## 4.1 Dimension-free communication cost

In order to remove the dependence on the dimension $d$ in the communication cost $b = \tilde{\Omega}_\delta\left(\max\left(n\varepsilon^2, \sqrt{d}\varepsilon\right)\right)$ from the previous section, we need to improve the performance of our scheme in the *small-sample* regime $n\varepsilon^2 = o(\sqrt{d}\varepsilon)$. Equivalently, we want to be able to achieve the centralized DP performance by using only $b = \tilde{\Omega}_\delta\left(n\varepsilon^2\right)$ bits per client when $n\varepsilon = o(\sqrt{d})$. Assuming $\varepsilon \approx 1$, note that this implies that the total communication bandwidth of the system $nb = o(d)$, i.e. the server can receive information about at most $nb = n^2\varepsilon^2 = o(d)$ coordinates. We show that in this regime the performance of the scheme can be improved by a priori restricting the server's attention to a subset of the coordinates.

We make the following modification to Algorithm 1: before performing Algorithm 1, the server randomly selects $d' \approx O\left(\min(d, n^2\varepsilon^2)\right)$ coordinates and only requires clients to run Algorithm 1 on them. We present the modified scheme in Algorithm 2 and summarize its performance in Theorem 4.4.

Similarly, we can obtain the following $\ell_2$ mean estimation via Kashin's representations:

**Theorem 4.4** ($\ell_2$ mean estimation.). *Let $x_1, ..., x_n \in \mathcal{B}_2(C)$ (i.e., $\|x_i\|_2 \le C$ for all $i \in [n]$), $\varepsilon, \delta > 0$, and $d' = \min\left(d, nb, \frac{n^2\varepsilon^2}{(\log(1/\delta) + \varepsilon)\log(d/\delta)}\right)$. Then there exists a*

$$\sigma^2 = O\left(\frac{C^2 \log(1/\delta)}{d'n^2\gamma^2} + \frac{C^2 d'(\log(1/\delta) + \varepsilon)\log(d'/\delta)}{dn^2\varepsilon^2}\right). \tag{5}$$

*such that Algorithm 2 is $(\varepsilon, \delta)$-DP. In addition, the (average) per-client communication cost is $\gamma d = b$ bits, and the $\ell_2^2$ estimation error is at most*

$$O\left(\max\left(\frac{C^2 d\log(d/\delta)}{nb}, \frac{C^2 d\log(d/\delta)(\log(1/\delta) + \varepsilon)}{n^2\varepsilon^2}\right)\right). \tag{6}$$

**Corollary 4.5.** *As long as* $b = \Omega\left(\frac{n\varepsilon^2}{\log(1/\delta)+\varepsilon}\right)$, *the* $\ell_2^2$ *error of mean estimation is*

$$O\left(\frac{C^2 d \log(d/\delta)(\log(1/\delta)+\varepsilon)}{n^2\varepsilon^2}\right).$$

As suggested by Corollary 4.5, we see that when $\varepsilon = O(1)$, $b = \tilde{\Omega}\left(n\varepsilon^2\right)$ bits per client are sufficient to achieve the order-optimal $\tilde{O}_\delta\left(\frac{c^2 d}{n^2\varepsilon^2}\right)$ error (even in the small sample regime $n \leq \sqrt{d}$), i.e. the communication cost of the scheme is independent of the dimension $d$.

### 4.2 Lower bounds

In this section, we argue that the estimation error in Theorem 4.4 is optimal up to an $\log(d/\delta)$ factor. Specifically, Theorem 5.3 of [31] shows that any $b$-bit *unbiased* compression scheme will incur $\Omega\left(\frac{C^2 d}{nb}\right)$ error for the $\ell_2$ mean estimation problem (even when privacy is not required). This matches the first term in (6) up to a logarithmic factor.

On the other hand, the centralized Gaussian mechanism (under a central $(\varepsilon,\delta)$-DP) achieves $O\left(\frac{C^2 d \log(1/\delta)}{n^2\varepsilon^2}\right)$ MSE [15] (which is order-optimal in most parameter regimes; see the lower bounds in Theorem 3.1 of [28] or Proposition 23 of [30]). Hence, we can conclude that

---

**Algorithm 2** CSGM with Coordinate Pre-selection

**Input:** users' data $x_1, ..., x_n$, coordinate selection $d' \leq d$, sampling parameters $\gamma := b/d'$, DP parameters $(\varepsilon,\delta)$.
**Output:** mean estimator $\hat{\mu}$.
Randomly select $d'$ coordinates $\mathcal{J} := \{j_1, ..., j_{d'}\} \subset [d]$.
**for** user $i \in [n]$ **do**
    Pre-processing $x_i$ by restricting it on $\mathcal{J}$:
    $x_i(\mathcal{J}) := (x_i(j_1), ..., x_i(j_{|\mathcal{J}|}))$.
**end for**
Apply CSGM (Algorithm 1) on $x_i(\mathcal{J}), i \in [n]$:
$\hat{\mu}_{\mathcal{J}} \leftarrow \mathsf{CSGM}\left(x_i(\mathcal{J}), i \in [n]\right)$.
**for** $j \in [d]$ **do**
    **if** $j \in \mathcal{J}$ **then**
        $\hat{\mu}_j = \hat{\mu}_{\mathcal{J}}(j)$.
    **else**
        $\hat{\mu}_j = 0$.
    **end if**
**end for**
**Return:** $\hat{\mu} := \left(\frac{d}{d'}\hat{\mu}_1, \frac{d}{d'}\hat{\mu}_2, ..., \frac{d}{d'}\hat{\mu}_d\right)$.

---

the total communication received by the server has to be at least $\Omega(n^2\varepsilon^2)$ bits in order to achieve the same error as the Gaussian mechanism. Therefore, the (average) per-client communication cost has to be at least $\Omega(n\varepsilon^2)$ bits. Hence we conclude that Algorithm 2 is optimal (up to a logarithmic factor).

For completeness, we state the communication lower bound in the following theorem:

**Theorem 4.6** (Communication lower bound for mean estimation under central DP). *Let* $x_1, ..., x_n \in \mathcal{B}_2(C)$. *Let* $Y_1, ..., Y_n$ *be any $b$-bit local reports generated from a (possibly interactive) compressor and be unbiased in the sense that* $\mathbb{E}\left[\sum_i Y_i\right] = \sum_i x_i$. *Then if* $\mathbb{E}\left[\left\|\frac{1}{n}\sum_i Y_i - \frac{1}{n}\sum_i x_i\right\|_2^2\right] \leq O\left(\frac{C^2 d \log(1/\delta)}{n^2\varepsilon^2}\right)$, *it holds that* $b = \Omega\left(\frac{n\varepsilon^2}{\log(1/\delta)}\right)$.

Finally, we remark that the logarithmic gap between the upper and lower bounds may be due to the specific composition theorem (Theorem III.3 of [39]) we use in our proof, which is simpler to work with but possibly slightly weaker. However, in our experiments, we compute and account for all privacy budgets with Rényi DP [69, 82], and hence can obtain better constants compared to our theoretical analysis.

## 5 Achieving the Optimal Trade-off via Shuffling

In Section 4 and Section A, we see that the communication cost can be reduced to $(\tilde{O}\left(n\varepsilon^2\right)$ for mean estimation and $\tilde{O}\left(\log\left(\lceil n\varepsilon^2 \rceil\right)\right)$ for frequency estimation) while still achieving the order-wise optimal error, as long as the server is *trusted*. On the other hand, when the server is untrusted, [33, 31] show that optimal error under $(\varepsilon,\delta)$-DP can be achieved with secure aggregation. However, the communication cost of these schemes is $\tilde{O}\left(n^2\varepsilon^2\right)$ bits per client for mean estimation and $\tilde{O}\left(n\varepsilon\right)$ bits per client for frequency estimation. This corresponds to a factor of $n$ increase for mean estimation and an exponential increase for frequency estimation. In this section, we investigate whether the

optimal communication-accuracy-privacy trade-off from the previous sections can be achieved when the server is not fully trusted.

In this section, we show that if there exists a *secure* shuffler that randomly permutes clients' locally privatized messages and releases them to the server, we can achieve the nearly optimal (within a $\log d$ factor) central-DP error in mean estimation with $\tilde{O}\left(n\varepsilon^2\right)$ bits of communication. We note that a similar result has been proven in a concurrent work [54]. Specifically, we present a mean estimation scheme that combines a local-DP mechanism with privacy amplification via shuffling by building on the following recent result [40, 43]:

**Lemma 5.1** ([43]). *Let $\mathcal{M}_i$ be an independent $(\varepsilon_0, 0)$-LDP mechanism for each $i \in [n]$ with $\varepsilon_0 \leq 1$ and $\pi$ be a random permutation of $[n]$. Then for any $\delta \in [0, 1]$ such that $\varepsilon_0 \leq \log\left(\frac{n}{16\log(2/\delta)}\right)$, the mechanism $\mathcal{S} : (x_1, \ldots, x_n) \mapsto \left(\mathcal{M}_1\left(x_{\pi(1)}\right), \ldots, \mathcal{M}_n\left(x_{\pi(n)}\right)\right)$ is $(\varepsilon, \delta)$-DP for some $\varepsilon$ such that $\varepsilon = O\left(\varepsilon_0 \frac{\sqrt{\log(1/\delta)}}{\sqrt{n}}\right)$.*

**Privacy analysis.** With the above amplification lemma, we only need to design the local randomizers $\mathcal{M}_i$ that satisfy $\varepsilon_0$-LDP. Note that the above lemma is only tight when $\varepsilon_0 = O(1)$, thus restricting the (amplified) central $\varepsilon = O(1/\sqrt{n})$, i.e. to be very small. To accommodate larger $\varepsilon$, users can send different portions of their messages to the server in separate shuffling rounds. Equivalently, we repeat the shuffled LDP mechanism for $T = O\left(\lceil n\varepsilon^2\rceil\right)$ rounds while ensuring that in each round, clients communicate an independent piece of information about their sample to the server. More precisely, within each round, each client applies the local randomizers $\mathcal{M}_i$ with a per-round *local privacy budget* $\varepsilon_0 = O(1)$ and sends an independent message to the server. This results in (amplified) central $O(1/\sqrt{n})$-DP per round, which after composition over $T = O\left(\lceil n\varepsilon^2\rceil\right)$ rounds leads to $\varepsilon$-DP for the overall scheme as suggested by the composition theorem [58]). We detail the algorithm in Algorithm 4 in Appendix F.

**Communication costs.** The communication cost of the above $T$-round scheme can be computed as follows. As shown in [32], the optimal communication cost of an $\varepsilon_0$-LDP mean estimation is $O\left(\lceil\varepsilon_0\rceil\right)$ bits. In addition, the (private-coin) SQKR scheme proposed in [32] uses $O\left(\lceil\varepsilon_0\rceil\log d\right)$ bits of communication (we state the formal performance guarantee for this scheme in Lemma 5.2), where compression is done by subsampling coordinates and privatization is performed with Randomized Response. Therefore, since the per-round $\varepsilon_0 = O(1)$, the total per-client communication cost is $O\left(n\varepsilon^2\log d\right)$, matching the optimal communication bounds in Section 4 within a $\log d$ factor.

**Lemma 5.2** (SQKR [32]). *For all $\varepsilon_0 > 0, b_0 > 0$, there exists a $(\varepsilon_0, 0)$-LDP mechanism using $b_0\log(d)$ bits such that $\hat{\mu}$ is unbiased and satisfies $\mathbb{E}\left[\|\mu\left(x^n\right) - \hat{\mu}\left(x^n\right)\|_2^2\right] = O\left(\frac{c^2 d}{n\min\left(\varepsilon_0^2, \varepsilon_0, b_0\right)}\right)$.*

Finally, we summarize the performance guarantee for the overall scheme (Algorithm 4) in the following theorem.

**Theorem 5.3** ($\ell_2$ mean estimation). *Let $x_1, ..., x_n \in \mathcal{B}_2(C)$ (i.e., $\|x_i\|_2 \leq C$ for all $i \in [n]$). For all $\varepsilon > 0, b > 0, n > 30$, and $\delta \in (\delta_{\min}, 1]$ where $\delta_{\min} = O\left(\frac{be^{-n}}{\log(d)}\right)$, Algorithm 4 combined with Kashin's representation and randomized rounding is $(\varepsilon, \delta)$-DP, uses no more than $b$ bits of communication, and achieves*

$$\mathbb{E}\left[\|\mu\left(x^n\right) - \hat{\mu}\left(x^n\right)\|_2^2\right] = O\left(C^2 d\max\left(\frac{\log(d)}{nb}, \frac{\log(b/\delta)(\log(1/\delta) + \varepsilon)}{n^2\varepsilon^2}\right)\right).$$

**Remark 5.4.** *As opposed to previous schemes Algorithm 1-3, the shuffled SQKR requires some condition on $\delta$, i.e., $\delta \in [\delta_{min}, 1]$ due to the specific shuffling lemma we used. In practice, however, $\delta_{\min}$ is small due to the exponential dependence on $n$. The order-wise optimal error of $O\left(\frac{C^2 d}{n^2\min(\varepsilon^2, \varepsilon)}\right)$ is achieved, up to logarithmic factors, when $b = \Omega_\delta\left(n\log(d)\min\left(\varepsilon^2, \varepsilon\right)\right)$.*

**Remark 5.5.** *We note that similar ideas of private mean estimation based on shuffling have been studied before, see, for instance, [53]. However, these papers do not use the above privacy budget splitting trick over multiple rounds, so their result is only optimal when $\varepsilon$ is very small. The above scheme can be viewed as a multi-message shuffling scheme [35, 51], and in particular, can be regarded as a generalization of the scalar mean estimation scheme [35] to $d$-dim mean estimation.*

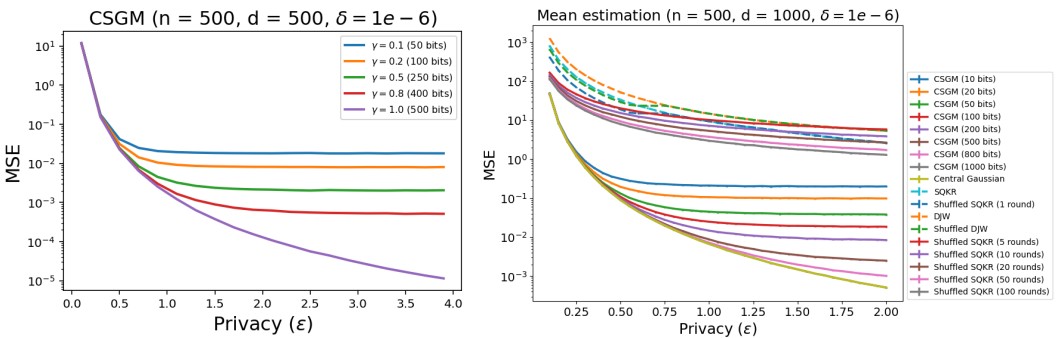

Figure 1: MSEs of CSGM (Algorithm 1) and shuffle LDP schemes.

# 6 Experiments

In this section, we empirically evaluate our mean estimation scheme (CSGM) from Section 4, examine its privacy-accuracy-communication trade-off, and compare it with other DP mechanisms (including the shuffling-based mechanism introduced in Section 5).

**Setup.** For a given dimension $d$, and number of samples $n$, we generate local vectors $X_i \in \mathbb{R}^d$ as follows: let $X_i(j) \overset{\text{i.i.d.}}{\sim} \frac{1}{\sqrt{d}} (2 \cdot \text{Ber}(0.8) - 1)$ where $\text{Ber}(0.8)$ is a Bernoulli random variable with bias $p = 0.8$. This ensures $\|X_i\|_\infty \leq 1/\sqrt{d}$ and $\|X_i\|_2 \leq 1$, and in addition, the empirical mean $\mu(X^n) \coloneqq \frac{1}{n} \sum_i X_i$ does not converge to 0. Note that as our goal is to construct an unbiased estimator, we did not project our final estimator back to the $\ell_\infty$ or $\ell_2$ space as the projection step may introduce bias. Therefore, the $\ell_2$ estimation error can be greater than 1. We account for the privacy budget with Rényi DP [69] and the privacy-amplification by subsampling lemma in [82] and convert Rényi DP to $(\varepsilon, \delta)$-DP via [30].

**Privacy-accuracy-communication trade-off of CSGM.** In the first experiment (left of Figure 1), we apply Algorithm 1 with different sampling rates $\gamma$, which leads to different communication budgets ($b = \gamma d$). Note that when $\gamma = 1$, the scheme reduces to the central Gaussian mechanism without compression. In Figure1, we see that with a fixed communication budget, CSGM approximates the central (uncompressed) Gaussian mechanism in the high privacy regime (small $\varepsilon$) and starts deviating from it when $\varepsilon$ exceeds a certain value. In addition, that value of $\varepsilon$ depends only on sample size $n$ and the communication budget $b$ and not the dimension $d$ as predicted by our theory: recall that the compression error dominates the total error, and hence the performance starts to deviate from the (uncompressed) Gaussian mechanism when $b = o(n\varepsilon^2)$, a condition that is independent of $d$. Observe, for example, that when $b = 50$ bits, the Gaussian mechanism starts outperforming CSGM at $\varepsilon \geq 0.5$ for both $d = 500$ and $d = 5000$. Hence, for $\varepsilon \approx 0.5$ CSGM is able to provide 10X compression when $d = 500$, but 100X compression when $d = 5000$ without impacting MSE.

**Comparison with local and shuffle DP.** Next, we compare the CSGM with local and shuffled DP for $d = 10^3$ and $n = 500$. For local DP, we consider the private-coin SQKR scheme introduced in Section 5 which uses $(\lceil \log d \rceil + 1) T = 11T$ bits for $T$ shuffling rounds and DJW [37] which is known to be order-optimal when $\varepsilon = O(1)$ (but is not communication-efficient). For shuffle-DP, we apply the amplification lemma in [43] to find the corresponding local $\varepsilon_0$ (see Section 5 for more details) and simulate both SQKR and DJW as the local randomizers.

The MSEs of all mechanisms are reported in the right of Figure 1. Our results suggest that for a fixed communication budget (say, 10 bits), the practical performance of CSGM significantly outperforms shuffled-DP mechanisms, including the shuffled SQKR and DJW, eventhough they have the same order-wise guarantees theoretically. In addition, the amplification gain of single-round shuffling diminishes fast as $\varepsilon$ increases. Indeed, when $\varepsilon \geq 0.8$, we observe no amplification gain compared to the pure local DP.

# 7 Limitations and Future Work

While we have characterized the (nearly) orderwise optimal trade-offs under central and shuffled DP, it remains unclear whether the *pre-constants* can be further sharpened, potentially enhancing the practical applicability of these approaches. Furthermore, our analysis is based on a classical privacy amplification technique. By employing more advanced accounting methods, it might be possible to shave off the logarithmic factors.

# 8 Acknowledgements

This work was supported in part by NSF Award # CCF-2213223.

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

# A  Distributed Frequency Estimation

In this section, we consider the frequency estimation problem for federated analytics. Recall that for the frequency estimation task, each client's private data $x_i \in \{0,1\}^d$ satisfies $\|x_i\|_0 = 1$, and the goal is to estimate $\pi := \frac{1}{n} \sum_i x_i$ by minimizing the $\ell_2$ (or $\ell_1, \ell_\infty$) error $\mathbb{E}\left[\|\pi - \hat{\pi}(Y^n)\|_2^2\right]$ subject to communication and $(\varepsilon, \delta)$-DP constraints. When the context is clear, we sometimes use $x_i$ to denote, by abuse of notation, the index of the item, i.e., $x_i \in [d]$.

To fully make use of the $\ell_0$ structure of the problem, a standard technique is applying a Hadamard transform to convert the $\ell_0$ geometry to an $\ell_\infty$ one and then leveraging the recursive structure of Hadamard matrices to efficiently compress local messages.

Specifically, for a given $b$-bit constraint, we partition each local item $x_i$ into $2^{b-1}$ chunks $x_i^{(1)}, ..., x_i^{(2^b-1)} \in \{0,1\}^B$, where $B := d/2^{b-1}$ and $x_i^{(j)} = x_i[B \cdot (j-1) : B \cdot j - 1]$. Note that since $x_i$ is one-hot, only one chunk of $x_i^{(j)}$ is non-zero. Then, client $i$ performs the following Hadamard transform for each chunk: $y_i^{(\ell)} = H_B \cdot x_i^{(\ell)}$, where $H_B$ is defined recursively as follows:

$$H_{2^n} = \frac{1}{\sqrt{2}} \begin{bmatrix} H_{2^{n-1}}, & H_{2^{n-1}} \\ H_{2^{n-1}}, & -H_{2^{n-1}} \end{bmatrix}, \text{ and } H_0 = [1].$$

Each client then generates a sampling vector $Z_{ij} \overset{\text{i.i.d.}}{\sim} \text{Bern}\left(\frac{1}{B}\right)$ via shared randomness that is also known by the server, and commits $(y_i^{(1)}(j), ..., y_i^{(2^{b-1})}(j))$ as its local report. Since $(y_i^{(1)}(j), ..., y_i^{(2^{b-1})}(j))$ only contains a single non-zero entry that can be $\frac{1}{\sqrt{B}}$ or $-\frac{1}{\sqrt{B}}$, the local report can be represented in $b$ bits ($b-1$ bits for the location of the non-zero entry and 1 bit for its sign).

From the local reports, the server can compute an unbiased estimator by summing them together (with proper normalization) and performing an inverse Hadamard transform. Moreover, with an adequate injection of Gaussian noise, the frequency estimator satisfies $(\varepsilon, \delta)$-DP.

The idea has been used in previous literature under local DP [19, 6, 3, 32], but in order to obtain the order-optimal trade-off under *central*-DP, one has to combine Hadamard transform with a random subsampling step and incorporate the privacy amplification due to random compression in the analysis. In Algorithm 3, we provide a summary of the resultant scheme which builds on the Recursive Hadamard Response (RHR) mechanism from [32], which was originally designed for communication-efficient frequency estimation under *local* DP.

In the following theorem, we control the $\ell_\infty$ error of Algorithm 3.

**Theorem A.1.** *Let $\hat{\pi}(x^n)$ be the output of Algorithm 3. Then it holds that for all $j \in [d]$,*

$$\mathbb{E}\left[|\pi(j) - \hat{\pi}(j)|\right] \leq \sqrt{\frac{\sum_i \mathbb{1}_{\{x_i \in [B \cdot (j-1) : B \cdot j - 1]\}}}{n^2} + \frac{\sigma^2}{B}}, \tag{7}$$

*and the $\ell_2^2$ and $\ell_1$ errors are bounded by*

$$\mathbb{E}\left[\|\pi - \hat{\pi}\|_2^2\right] \leq \frac{B}{n} + \frac{d\sigma^2}{B}, \text{ and} \tag{8}$$

$$\mathbb{E}\left[\|\pi - \hat{\pi}\|_1\right] \leq \sqrt{\frac{dB}{n} + \frac{d^2\sigma^2}{B}}. \tag{9}$$

**Theorem A.2.** *For any $\varepsilon, \delta > 0$, Algorithm 3 is $(\varepsilon, \delta)$-DP, if*

$$\sigma^2 \geq O\left(\frac{B^2 \log(B/\delta)}{n^2} + \frac{B(\log(1/\delta) + \varepsilon)\log(B/\delta)}{n^2\varepsilon^2}\right).$$

By combining Theorem A.1 and Theorem A.2, we conclude that Algorithm 3 achieves $(\varepsilon, \delta)$-DP with $\ell_2^2$ error

$$O\left(\frac{B}{n} + \frac{dB \log(B/\delta)}{n^2} + \frac{d(\log(1/\delta) + \varepsilon)\log(B/\delta)}{n^2\varepsilon^2}\right)$$

$$= O\left(\frac{d}{n2^b} + \frac{d^2 \log(d/\delta)}{n^2 2^b} + \frac{d(\log(1/\delta) + \varepsilon)\log(d/\delta)}{n^2\varepsilon^2}\right).$$

**Algorithm 3** Subsampled Recursive Hadamard Response

---

**Input:** user data $x_1, ..., x_n \in \{0,1\}^d$ (where $d$ is a power of two), DP parameters $(\varepsilon, \delta)$, communication budget $b$.

**Output:** frequency estimate $\hat{\pi}$

Set $B := d/2^{b-1}$ and partition each one-hot vector $x_i$ into $2^{b-1}$ chunks: $x_i^{(1)}, ..., x_i^{(2^b-1)} \in \{0,1\}^B$.

**for** user $i \in [n]$ **do**

    Compute the Hadamard transform of each chunk: $y_i^{(\ell)} = H_B \cdot x_i^{(\ell)}$.

    **for** coordinate $j \in [B]$ **do**

        Draw $Z_{i,j} \overset{\text{i.i.d.}}{\sim} \text{Bern}\left(\frac{1}{B}\right)$

        **if** $Z_{i,j} = 1$ **then**

            Send $(y_i^{(1)}(j), ..., y_i^{(2^{b-1})}(j))$ to the server.

        **end if**

    **end for**

**end for**

Server computes the average: $\forall \ell \in [2^{b-1}], j \in [B]$,

$$\hat{y}^{(\ell)}(j) := \frac{B}{n} \sum_{i:Z_{ij}=1} y_i^{(\ell)}(j) + N(0, \sigma^2),$$

where $\sigma^2$ is computed according to Theorem A.2.

Server performs the inverse Hadamard transform $\hat{\pi}^{(\ell)} = H_B \cdot \hat{y}^{(\ell)}$, for $\ell = 1, ..., B$.

**Return:** $\hat{\pi} = \left( \left( \hat{\pi}^{(1)} \right)^{\mathsf{T}}, ..., \left( \hat{\pi}^{(2^{b-1})} \right)^{\mathsf{T}} \right)$.

---

Notice that when $n = \tilde{\Omega}(d)$, the error can be simplified to

$$O\left( \frac{d}{n2^b} + \frac{d(\log(1/\delta) + \varepsilon)\log(d/\delta)}{n^2\varepsilon^2} \right),$$

which matches the order-optimal estimation error (up to a $\log d$ factor) subject to a $b$-bit constraint [55, 3, 2] and $(\varepsilon, \delta)$-DP constraint [15, 7].

# B Proof of Theorem 4.1

It is trivial to see that the average communication cost is $d \cdot \gamma = b$ bits. To compute the $\ell_2^2$ estimation error, observe that

$$\mathbb{E}\left[\|\hat{\mu}_{x^n} - \mu_{x^n}\|_2^2\right]$$

$$= \sum_{j=1}^{d} \mathbb{E}\left[\left(\frac{1}{n\gamma}\sum_i x_i(j) \cdot Z_{i,j} + N(0,\sigma^2) - \frac{1}{n}\sum_i x_i(j)\right)^2\right]$$

$$= \sum_{j=1}^{d} \frac{1}{n^2}\mathbb{E}\left[\left(\frac{1}{\gamma}\sum_i x_i(j) \cdot Z_{i,j} - \sum_i x_i(j)\right)^2\right] + d\sigma^2$$

$$= \sum_{j=1}^{d} \frac{1}{n^2}\mathbb{E}\left[\left(\frac{1}{\gamma}\sum_i x_i(j) \cdot Z_{i,j}\right)^2\right] - \frac{1}{n^2}\left(\sum_i x_i(j)\right)^2 + d\sigma^2$$

$$= \sum_{j=1}^{d} \frac{1}{n^2}\mathbb{E}\left[\frac{1}{\gamma^2}\sum_i x_i^2(j) \cdot Z_{i,j}^2 + \frac{1}{\gamma^2}\sum_{i\neq i'} x_i(j)x_{i'}(j)Z_{i,j}Z_{i',j}\right] - \frac{1}{n^2}\left(\sum_i x_i(j)\right)^2 + d\sigma^2$$

$$= \sum_{j=1}^{d} \frac{1}{n^2}\left(\frac{1}{\gamma}\sum_i x_i^2(j) + \sum_{i\neq i'} x_i(j)x_{i'}(j)\right) - \frac{1}{n^2}\left(\sum_i x_i(j)\right)^2 + d\sigma^2$$

$$= \sum_{j=1}^{d} \frac{1}{n^2}\left(\frac{1}{\gamma} - 1\right)\left(\sum_i x_i^2(j)\right) + d\sigma^2$$

$$\leq \frac{dc^2}{n\gamma} + d\sigma^2,$$

which yields the inequality of (2). Next, we analyze the privacy of Algorithm 1. We first the following two lemmas for subsampling and the Gaussian mechanism:

**Lemma B.1** ([66, 82]). *If $\mathcal{M}$ is $(\varepsilon, \delta)$-DP, then $\mathcal{M}'$ that applies $\mathcal{M} \circ$ PoissonSample satisfies $(\varepsilon', \delta')$-DP with $\varepsilon' = \log(1 + \gamma(e^\varepsilon - 1))$ and $\delta' = \gamma\delta$.*

**Lemma B.2** ([15]). *For any $\varepsilon, \delta \in (0,1)$, the Gaussian output perturbation mechanism with $\sigma^2 := \frac{\Delta^2 2\log(1.25/\delta)}{\varepsilon^2}$ satisfies $(\varepsilon, \delta)$-DP, where $\Delta$ is the $\ell_2$ sensitivity of the target function.*

Now, we use the above two lemmas to analyze the per-coordinate privacy leakage of Algorithm 1. For simplicity, we analyze the sum of $x_i(j)$'s instead (and normalized it in the last step). Let $S_j(x^n) := \sum_{i=1}^n (x_i(j))$, then clearly the sensitivity of $S_j(x^n)$ is $2c$, so Lemma B.2 implies $S_j(x^n) + N(0, \sigma_1^2)$ satisfies $(\varepsilon_1, \delta_1)$-DP if we set $\sigma_1^2 = \frac{2c^2 \log(1.25/\delta_1)}{\varepsilon_1^2}$ (assuming $\varepsilon_1 < 1$). Next, if applying subsampling before computing the sum, i.e.,

$$S_j \circ \mathsf{PoissonSample}_\gamma(x^n) := \sum_{i=1}^n x_i(j)Z_{i,j},$$

where $Z_{i,j} \overset{\text{i.i.d.}}{\sim} \mathrm{Bern}(\gamma)$ as defined in Algorithm 1, then by Lemma B.1,

$$S_j \circ \mathsf{PoissonSample}_\gamma(x^n) + N(0, \sigma_1^2)$$

satisfies $(\varepsilon_2, \delta_2)$-DP with $\varepsilon_2 := \log(1 + \gamma(e^{\varepsilon_1} - 1)) = C_1\gamma\varepsilon_1$ (since we assume $\epsilon_1 < 1$) and $\delta_2 := \gamma\delta_1$. Equivalently, we have

$$\begin{cases} \varepsilon_1 = \tilde{C}_1 \frac{1}{\gamma}\varepsilon_2 \\ \delta_1 = \frac{1}{\gamma}\delta_2. \end{cases} \tag{10}$$

Now, since we have established the per-coordinate privacy leakage, we apply the following composition theorem to account for the total privacy budgets.

**Theorem B.3** ([39]). *For any $\varepsilon > 0$, $\delta \in [0, 1]$ and $\tilde{\delta} \in (0, 1]$, the class of $(\varepsilon, \delta)$-DP mechanisms satisfies $(\tilde{\varepsilon}_{\tilde{\delta}}, d\delta + \tilde{\delta})$-DP under $d$-fold adaptive composition, for*

$$\tilde{\varepsilon}_{\tilde{\delta}} = d\varepsilon(e^\varepsilon - 1) + \varepsilon\sqrt{2d\log(1/\tilde{\delta})}.$$

According Theorem B.3, Algorithm 1 satisfies $(\varepsilon, \delta)$-DP for

$$\varepsilon = d\varepsilon_2(e^{\varepsilon_2} - 1) + \varepsilon_2\sqrt{2d\log(1/\tilde{\delta})}, \tag{11}$$

and $\delta = d\delta_2 + \tilde{\delta}$ (where $\tilde{\delta}$ is a free parameter that we can optimize).

Consequently, for a pre-specified (total) privacy budget $(\varepsilon, \delta)$, we set parameters as follows. Let $\tilde{\delta} = \frac{\delta}{2}$ and $\delta_1 = \frac{1}{\gamma}\delta_2 = \frac{1}{2d\gamma}\delta$. Let $\varepsilon_2 \le 1$ so that $e^{\varepsilon_2} - 1 \le 2\varepsilon_2$ holds. Then (11) implies Algorithm 1 is

$$\varepsilon = 2d\varepsilon_2^2 + \varepsilon_2\sqrt{2d\log(1/\tilde{\delta})} \ge d\varepsilon_2(e^{\varepsilon_2} - 1) + \varepsilon_2\sqrt{2d\log(1/\tilde{\delta})}.$$

Solving the above quadratic (in-)equality for $\varepsilon_2$, it suffices that

$$\varepsilon_2 = \min\left(1, \frac{-\sqrt{2d\log(2/\delta)} + \sqrt{2d\log(2/\delta) + 8\varepsilon d}}{4d}\right) = O\left(\min\left(1, \frac{\varepsilon}{\sqrt{d(\log(1/\delta) + \varepsilon)}}\right)\right).$$

Consequently, we set $\varepsilon_1 = \frac{\tilde{C}_1}{\gamma}\varepsilon_2 = O\left(\min\left(1, \frac{\varepsilon}{\gamma\sqrt{d(\log(1/\delta)+\varepsilon)}}\right)\right)$ (note that we require $\varepsilon_1 = O(1)$ so that (10) holds).

Plugging $(\varepsilon_1, \delta_1)$ into $\sigma_1^2$, we need to choose

$$\sigma_1^2 := \frac{2c^2\log(1.25/\delta_1)}{\varepsilon_1^2} = \Omega\left(\max\left(c^2\log(d/\delta), \frac{\gamma^2 c^2 d(\log(1/\delta) + \varepsilon)\log(d/\delta)}{\varepsilon^2}\right)\right).$$

Finally, as we are interested in estimating the (subsampled) mean instead of the sum, we will normalize the private sum by

$$\hat{\mu}_j(x^n) = \frac{1}{n\gamma}\left(S_j \circ \mathsf{PoissonSample}_\gamma(x^n) + N(0, \sigma_1^2)\right) = \frac{1}{n\gamma}S_j \circ \mathsf{PoissonSample}_\gamma(x^n) + N(0, \sigma^2),$$

where

$$\sigma^2 = \Theta\left(\max\left(\frac{c^2\log(d/\delta)}{n^2\gamma^2}, \frac{c^2 d(\log(1/\delta) + \varepsilon)\log(d/\delta)}{n^2\varepsilon^2}\right)\right).$$

Plugging in $\sigma^2$ above and $\gamma = b/d$ yields the desired accuracy in Theorem 4.1. $\qquad\square$

Since we will reuse the above result, we summarize it into the following lemma:

**Lemma B.4.** *Let $f_i : \mathbb{R}^{d \times m} \mapsto \mathbb{R}^D$ for $i = 1, ..., B$ be $n$ functions with sensitivity bounded by $\Delta$ (where the number of inputs $m$ can be a random variable). Then*

$$\left(f_1 \circ \mathsf{PoissonSample}_\gamma(x^n) + N(0, \sigma^2), ..., f_B \circ \mathsf{PoissonSample}_\gamma(x^n) + N(0, \sigma^2)\right)$$

*satisfies $(\varepsilon, \delta)$-DP, if*

$$\sigma^2 \ge O\left(\max\left(\Delta^2\log(B/\delta), \frac{\gamma^2\Delta^2 B(\log(1/\delta) + \varepsilon)\log(B/\delta)}{\varepsilon^2}\right)\right).$$

## C  Omitted details of dimension-free communication cost

### C.1  Proof of Theorem 4.4

To prove Theorem 4.4, it suffices to prove the following $\ell_\infty$ version:

**Theorem C.1.** *Let* $x_1, ..., x_n \in \{-c, c\}^d$, $d' = \min\left(nb, \frac{n^2\varepsilon^2}{(\log(1/\delta)+\varepsilon)\log(d/\delta)}\right)$, *and*

$$\sigma^2 = O\left(\frac{c^2\log(1/\delta)}{n^2\gamma^2} + \frac{c^2 d'\left(\log(d'/\delta)+\varepsilon\right)\log(d'/\delta)}{n^2\varepsilon^2}\right). \tag{12}$$

*Then Algorithm 2 is $(\varepsilon, \delta)$-DP and yields an unbiased estimator on $\mu$. In addition, the (average) per-client communication cost is $\gamma d' = b$ bits, and the $\ell_2^2$ estimation error is at most*

$$O\left(c^2 d^2 \log\left(\frac{d}{\delta}\right)\max\left(\frac{1}{nb}, \frac{(\log(1/\delta)+\varepsilon)}{n^2\varepsilon^2}\right)\right). \tag{13}$$

With a slight abuse of notation, we let $\mu_{\mathcal{J}} \in \mathbb{R}^d$ be such that

$$\mu_{\mathcal{J}}(j) = \begin{cases} 0, & \text{if } j \notin \mathcal{J} \\ \frac{d\mu_j}{d'}, & \text{else.} \end{cases}$$

Note that $\mu_{\mathcal{J}}$ is an unbiased estimate of $\mu$ if $\mathcal{J}$ is selected uniformly at random. Then the $\ell_2^2$ error can be controlled by

$$
\begin{aligned}
\mathbb{E}\left[\|\mu - \hat{\mu}\|_2^2\right] &\overset{(a)}{=} \mathbb{E}\left[\|\mu - \mu_{\mathcal{J}}\|_2^2\right] + \mathbb{E}\left[\|\mu_{\mathcal{J}} - \hat{\mu}\|_2^2\right] \\
&\overset{(b)}{\leq} \mathbb{E}\left[\|\mu - \mu_{\mathcal{J}}\|_2^2\right] + \frac{d^2}{d'^2}O\left(\max\left(\frac{d'^2 c^2}{nb}, \frac{d'^3 c^2 \log(d/\delta)}{n^2 b^2}, \frac{c^2 d'^2(\log(1/\delta)+\varepsilon)\log(d/\delta)}{n^2\varepsilon^2}\right)\right) \\
&= \mathbb{E}\left[\|\mu - \mu_{\mathcal{J}}\|_2^2\right] + O\left(\max\left(\frac{d^2 c^2}{nb}, \frac{d^2 d' c^2 \log(d/\delta)}{n^2 b^2}, \frac{c^2 d^2(\log(1/\delta)+\varepsilon)\log(d/\delta)}{n^2\varepsilon^2}\right)\right) \\
&\overset{(c)}{\leq} \frac{d^2 c^2}{d'} + O\left(\max\left(\frac{d^2 c^2}{nb}, \frac{d^2 d' c^2 \log(d/\delta)}{n^2 b^2}, \frac{c^2 d^2(\log(1/\delta)+\varepsilon)\log(d/\delta)}{n^2\varepsilon^2}\right)\right),
\end{aligned}
$$

where (a) holds since $\mu_{\mathcal{J}}$ is an unbiased estimate of $\mu$ and conditioned on $\mathcal{J}$, $\hat{\mu}$ is an unbiased estimate of $\mu_{\mathcal{J}}$; (b) follows from Theorem 4.1; (c) holds due to the following fact:

$$\mathbb{E}\left[\|\mu - \mu_{\mathcal{J}}\|_2^2\right] \leq \sum_{j\in\mathcal{J}}\mu_{\mathcal{J}}(j)^2 + \sum_{j\in[d]}\mu_j^2 \leq \frac{d^2 c^2}{d'} + dc^2 \leq \frac{2d^2 c^2}{d'}.$$

Therefore, by setting $d' = \min\left(nb, \frac{n^2\varepsilon^2}{(\log(1/\delta)+\varepsilon)\log(d/\delta)}\right)$ we ensure the first term in (c) is always smaller than the second term, and the second term can be simplified as follows:

$$
\begin{aligned}
&O\left(c^2 d^2 \max\left(\frac{1}{nb}, \frac{d'\log(d/\delta)}{n^2 b^2}, \frac{(\log(1/\delta)+\varepsilon)\log(d/\delta)}{n^2\varepsilon^2}\right)\right) \\
&\leq O\left(c^2 d^2 \max\left(\frac{1}{nb}, \frac{nb\log(d/\delta)}{n^2 b^2}, \frac{(\log(1/\delta)+\varepsilon)\log(d/\delta)}{n^2\varepsilon^2}\right)\right) \\
&\leq O\left(c^2 d^2 \log(d/\delta) \max\left(\frac{1}{nb}, \frac{(\log(1/\delta)+\varepsilon)}{n^2\varepsilon^2}\right)\right).
\end{aligned}
$$

Finally, applying the same trick of Kashin's representation, we can transform the $\ell_\infty$ geometry to $\ell_2$ (similar to Corollary 4.3), hence proving Theorem 4.4. $\qquad\square$

## D    Proof of Theorem A.1

Let $\pi := \frac{1}{n}\sum_i x_i$ and $\pi^{(\ell)}$ be defined in the same way as $x_i^{(\ell)}$ for $\ell \in [B]$. Then our goal is to bound $\left|\pi^{(\ell)}(j) - \hat{\pi}^{(\ell)}(j)\right|$, for all $\ell \in [2^{b-1}]$ and $j \in [B]$.

To this end, let $y^{(\ell)} := H_B \cdot \pi^{(\ell)}$ (so it holds that $\pi^{(\ell)} = \frac{1}{B}H_B \cdot y^{(\ell)}$). Then we have

$$
\begin{aligned}
\mathbb{E}\left[\left|\pi^{(\ell)}(j) - \hat{\pi}^{(\ell)}(j)\right|\right] &\overset{(a)}{\leq} \sqrt{\mathbb{E}\left[\left(\pi^{(\ell)}(j) - \hat{\pi}^{(\ell)}(j)\right)^2\right]} \\
&= \sqrt{\mathbb{E}\left[\left(\frac{1}{B}H_B \cdot \left(y^{(\ell)} - \hat{y}^{(\ell)}\right)(j)\right)^2\right]}. \tag{14}
\end{aligned}
$$

Next, observe that due to the subsampling step, for all $\ell \in [2^{b-1}]$ and $j \in [B]$,

$$\hat{y}^{(\ell)}(j) = \frac{B}{n} \sum_{i=1}^{n} \langle (H_B)_j, x_i^{(\ell)} \rangle \cdot Z_{ij} + N(0, \sigma^2),$$

where recall that $Z_{ij} \overset{\text{i.i.d.}}{\sim} \text{Ber}(1/B)$. Therefore, $\hat{y}^{(\ell)}(j)$ is an unbiased estimator of $y^{(\ell)}(j)$. In addition, since we choose $Z_{ij}$ independently in Algorithm 3, $\hat{y}^{(\ell)}(j)$'s are independent for different $j$'s, so we have

$$\mathbb{E}\left[\left(\hat{y}^{(\ell)}(j) - y^{(\ell)}(j)\right)^2\right] = \text{Var}\left(\hat{y}^{(\ell)}(j)\right)$$

$$= \sigma^2 + \frac{B^2}{n^2} \sum_{i=1}^{n} \langle (H_B)_j, x_i^{(\ell)} \rangle^2 \text{Var}(Z_{ij})$$

$$\leq \sigma^2 + \frac{B}{n^2} \sum_{i=1}^{n} \langle (H_B)_j, x_i^{(\ell)} \rangle^2$$

$$= \sigma^2 + \frac{B}{n^2} \underbrace{\sum_{i=1}^{n} \mathbb{1}_{\{x_i \in \ell\text{-th chunk}\}}}_{:=C_\ell}, \tag{15}$$

and for all $j \neq j'$

$$\mathbb{E}\left[\left(\hat{y}^{(\ell)}(j) - y^{(\ell)}(j)\right) \cdot \left(\hat{y}^{(\ell)}(j') - y^{(\ell)}(j')\right)\right] = 0. \tag{16}$$

Therefore, we continue bounding (14) as follows:

$$\sqrt{\mathbb{E}\left[\left(\frac{1}{B} H_B \cdot \left(y^{(\ell)} - \hat{y}^{(\ell)}\right)(j)\right)^2\right]} = \sqrt{\frac{1}{B^2} \mathbb{E}\left[\langle (H_B)_j, \left(\hat{y}^{(\ell)} - y^{(\ell)}\right)\rangle^2\right]}$$

$$= \sqrt{\frac{1}{B^2} \mathbb{E}\left[\left(\sum_{k=1}^{B} (H_B)_{jk} \cdot \left(\hat{y}^{(\ell)}(k) - y^{(\ell)}(k)\right)\right)^2\right]}$$

$$\overset{(a)}{=} \sqrt{\frac{1}{B^2} \mathbb{E}\left[\sum_{k=1}^{B} \left(\hat{y}^{(\ell)}(k) - y^{(\ell)}(k)\right)^2\right]}$$

$$\overset{(b)}{=} \sqrt{\frac{C_\ell}{n^2} + \frac{\sigma^2}{B}}$$

$$\overset{(c)}{\leq} \sqrt{\frac{1}{n} + \frac{\sigma^2}{B}},$$

where (a) holds since each entry of $H_B$ takes value in $\{-1, 1\}$ and by (16), (b) holds due to (15), and (c) holds because $C_\ell \leq n$ for all $\ell$.

Finally, to bound the $\ell_2^2$ error, observe that the above analysis ensures that

$$\mathbb{E}\left[\left(\pi^{(\ell)}(j) - \hat{\pi}^{(\ell)}(j)\right)^2\right] \leq \frac{C_{\ell(j)}}{n^2} + \frac{\sigma^2}{B},$$

where $\ell(j) \in [2^{b-1}]$ is the index of the chuck containing $j$. Therefore, summing over $j \in [d]$, we must have

$$\mathbb{E}\left[\left\|\pi^{(\ell)} - \hat{\pi}^{(\ell)}\right\|_2^2\right] \leq \sum_{j=1}^{d} \frac{C_{\ell(j)}}{n^2} + \frac{d\sigma^2}{B} = \frac{B}{n} + \frac{d\sigma^2}{B},$$

since

$$\sum_{j} C_{\ell(j)} = \sum_{\ell=1}^{2^{b-1}} \sum_{j' \in \ell\text{-th chunk}} \sum_{i=1}^{n} \mathbb{1}_{\{i \in \ell - \text{th chunk}\}} = B \sum_{\ell=1}^{2^{b-1}} \sum_{i=1}^{n} \mathbb{1}_{\{i \in \ell - \text{th chunk}\}} = B \cdot n.$$

$\square$

# E  Proof of Theorem A.2

Let $f_j(x^n) := (\pi^{(1)}(j), ..., \pi^{(2^{b-1})}(j))$, for $j = 1, ..., B$. Then the $\ell_2$ sensitivity of $f_j$ is $\Delta = \frac{B}{n}$. Set the sampling rate $\gamma = \frac{1}{B}$ and the proof is complete by Lemma B.4. $\qquad\square$

# F  Algorithm of Shuffled SQKR

---
**Algorithm 4** Shuffled SQKR
---
**Input:** users' data $x_1, \ldots, x_n$, local-DP parameter $\varepsilon_0$, communication parameters $b_0, T$
**Output:** mean estimator $\hat{\mu}$
**for** round $k \in [T]$ **do**
    **for** user $i \in [n]$ **do**
        Sample $s(i,1), \ldots, s(i, b_0) \overset{\text{i.i.d.}}{\sim} \mathsf{Unif}[d]$
        Sample $Z \sim \mathsf{Bern}\left(\frac{e^{\varepsilon_0}}{e^{\varepsilon_0} + 2^{b_0} - 1}\right)$
        **if** Z=1 **then**
            Set $Y(i,1), \ldots, Y(i, b_0) \leftarrow x_i(s(i,1)), \ldots, x_i(s(i, b_0))$
        **else**
            Sample $Y(i,1), \ldots, Y(i, b_0) \overset{\text{i.i.d.}}{\sim} \mathsf{Unif}\{-c, c\}$
        **end if**
        Send $Y(i,1), \ldots, Y(i, b_0)$ and $s(i,1), \ldots, s(i, b_0)$ to shuffler
    **end for**
    Shuffler samples a permutation $\pi \sim \mathsf{Unif}\{f : [n] \to [n] \text{ bijective}\}$
    **for** $j \in [b_0]$ **do**
        Shuffler sends $Y(\pi(1), j), \ldots, Y(\pi(n), j)$ and $s(\pi(1), j), \ldots, s(\pi(n), j)$ to server
    **end for**
    $\hat{\mu}^{(k)} \leftarrow \frac{d}{n b_0} \frac{e^{\varepsilon_0} + 2^{b_0} - 1}{e^{\varepsilon_0} - 1} \sum_{i=1}^{n} \sum_{j=1}^{b_0} Y(\pi(i), j) e_{s(\pi(i), j)}$
**end for**
Return $\hat{\mu} := \frac{1}{T} \sum_{k=1}^{T} \hat{\mu}^{(k)}$
---

# G  Proof of Theorem 5.3

Each round $x^n \mapsto \hat{\mu}^{(k)}$ of Algorithm 4 implements the private-coin SQKR scheme of [32], achieving the communication cost and error as stated in Lemma 5.2.

**Lemma G.1** (SQKR [32]). *For all $\varepsilon_0 > 0, b_0 > 0$, the random mapping $x_i \mapsto Y(i,1), \ldots, Y(i, b_0), s(i,1), \ldots, s(i, b_0)$ in Algorithm 4 is $(\varepsilon_0, 0)$-LDP and has output that can be communicated in $b_0 \log(d)$ bits, and the $\hat{\mu}^{(k)}$ computed from $Y(i,1), \ldots, Y(i, b_0), s(i,1), \ldots, s(i, b_0)$ is an unbiased estimator of $\mu$ satisfying*

$$\max_{x^n} \mathbb{E}\left[\left\|\mu(x^n) - \hat{\mu}^{(k)}(x^n)\right\|_2^2\right] = O\left(\frac{C^2 d}{n \min(\varepsilon_0^2, \varepsilon_0, b_0)}\right). \tag{17}$$

We now characterize the error performance of Algorithm 4 for general choices of parameters that satisfy privacy and communication constraints.

**Proposition G.2.** *For all $\varepsilon > 0, b > 0, n > 0$, with any arbitrary choice of*

$$\delta_1 \in \left(e^{-n/16e}, 1\right] \tag{18}$$

$$\delta_2 \in (0, 1], \tag{19}$$

*there exists a choice of parameters $\varepsilon_0, b_0, T$ such that Algorithm 4 is $(\varepsilon, T\delta_1 + \delta_2)$-DP, uses no more than $b$ bits of communication, and produces $\hat{\mu}$ such that*

$$\max_{x^n} \mathbb{E}\left[\|\mu - \hat{\mu}\|_2^2\right] = O\left(\max\left(\frac{C^2 d \log(d) b_0}{nb}, \frac{C^2 d \log(1/\delta_1)(\log(1/\delta_2) + \varepsilon)}{n^2 \varepsilon^2}\right)\right). \tag{20}$$

*Proof.* For arbitrary choice of

$$b_0 < \log\left(\frac{n}{16\log(2)}\right), \tag{21}$$

it suffices to choose

$$T = \left\lfloor \frac{b}{(\log_2(d)+1)b_0} \right\rfloor \tag{22}$$

$$\varepsilon_0 = O\left(\min\left(1, \frac{\varepsilon\sqrt{n}}{\sqrt{T\log(1/\delta_1)\left(\log(1/\delta_2)+\varepsilon\right)}}\right)\right). \tag{23}$$

The fact that Algorithm 4 uses less than $b$ bits is immediate from the choice of $T$.

Applying Lemma G.1, by construction the mapping from each $x_i$ to $Y(i,1),\ldots,Y(i,b_0)$ is $(\varepsilon_0,0)$-LDP. By assumption

$$\delta_1 > e^{-n/16e}, \tag{24}$$

the inequality

$$1 < \log\left(\frac{n}{16\log(2/\delta_1)}\right) \tag{25}$$

is satisfied. Then the choice of

$$\varepsilon_0 \leq 1 \tag{26}$$

also satisfies $\varepsilon_0 \leq \log\left(\frac{n}{16\log(2/\delta)}\right)$, so by Lemma 5.1 the mapping $x^n \mapsto \hat{\mu}^{(k)}$ is $(\varepsilon_1, \delta_1)$-DP, where

$$\varepsilon_1 = O\left(\frac{\varepsilon_0\sqrt{\log(1/\delta_1)}}{\sqrt{n}}\right). \tag{27}$$

Since the output of Algorithm 4 is a function of $\left(\hat{\mu}^{(1)},\ldots,\hat{\mu}^{(T)}\right)$, by B.3 it suffices to have

$$\varepsilon_1 = O\left(\min\left(1, \frac{\varepsilon}{\sqrt{T(\log(1/\delta_2)+\varepsilon)}}\right)\right) \tag{28}$$

for Algorithm 4 to be $(\varepsilon, T\delta_1 + \delta_2)$-DP. The first inequality follows from the assumption of $\delta_1 > e^{-n/16e}$ and choice of $\varepsilon_0 = O(1)$, and the second from choice of

$$\varepsilon_0 = O\left(\frac{\varepsilon\sqrt{n}}{\sqrt{T\log(1/\delta_1)\left(\log(1/\delta_2)+\varepsilon\right)}}\right). \tag{29}$$

Since $\varepsilon_0 \leq 1 \leq b$, we have $\min(\varepsilon_0^2, \varepsilon_0, b) = \varepsilon_0^2$. Applying Lemma G.1,

$$\max_{x^n} \mathbb{E}\left[\|\mu - \hat{\mu}\|_2^2\right] = \frac{1}{T}\max_{x^n}\mathbb{E}\left[\left\|\mu - \hat{\mu}^{(1)}\right\|_2^2\right] \tag{30}$$

$$= O\left(\frac{d}{Tn\varepsilon_0^2}\right) \tag{31}$$

$$= O\left(\max\left(\frac{d}{Tn}, \frac{d\log(1/\delta_1)\left(\log(1/\delta_2)+\varepsilon\right)}{n^2\varepsilon^2}\right)\right). \tag{32}$$

Substituting the choice of $T$ gives the desired result. $\qquad\square$

To show Theorem 5.3, it suffices to choose

$$b_0 = 1 \tag{33}$$

$$\delta_1 = \frac{\delta}{2T} \tag{34}$$

$$\delta_2 = \frac{\delta}{2}, \tag{35}$$

which requires $n > 16e\log(2) \approx 30.14$ due to (21), and apply the previous proposition.

# H   Using Kashin's Representation

We first introduce the idea of a tight frame in Kashin's representation. A tight frame is a set of vectors $\{u_j\}_{j=1}^D \subset \mathbb{R}^d$ that satisfy Parseval's identity, i.e. $\|x\|_2^2 = \sum_{j=1}^D \langle u_j, x \rangle^2$ for all $x \in \mathbb{R}^d$.

A frame can be viewed as a generalization of the notion of an orthogonal basis in $\mathbb{R}^d$ for $D > d$. To increase robustness, we wish the information to be spread evenly across different coefficients, which motivates the following definition of a Kashin's representation:

**Definition H.1.** *For a set of vectors $\{u_j\}_{j=1}^D$, we say the expansion*

$$x = \sum_{j=1}^D a_j u_j, \text{ with } \max_j |a_j| \leq \frac{K}{\sqrt{D}} \|x\|_2$$

*is a Kashin's representation of the vector $x$ at level $K$ .*

By Theorem 3.5 and Theorem 4.1 in [67], we have the following lemma:

**Lemma H.2.** *There exists a tight frame $U = [u_1, ..., u_D]$ with (1) $D = \Theta(d)$ and (2) level $K = O(1)$.*

The above lemma implies that for each $x_i \in \mathbb{R}^d$ such that $\|x_i\|_2 \leq 1$, one can always represent each $x_i$ with coefficients $y_i \in [-\gamma_0/\sqrt{d}, \gamma_0/\sqrt{d}]^{\gamma_1 d}$ for some $\gamma_0, \gamma_1 > 0$ and $x_i = Uy_i$.

# I   Rényi-DP for Shuffled SQKR

In this section we restate some results for RDP which are useful for privacy accounting in experiments.

Following the proof of Corollary 4.3 in [44], applying Theorem 4.1 in the same paper yields the following.

**Lemma I.1.** *Let $\mathcal{M}_i$ be an independent $(\varepsilon_0, 0)$-LDP mechanism for each $i \in [n]$ with $\varepsilon_0 \leq 1$ and $\pi$ be a random permutation of $[n]$. Then for any $\alpha < \frac{n}{16\varepsilon_0 \exp(\varepsilon_0)}$, the mechanism*

$$\mathcal{S} : (x_1, \ldots, x_n) \mapsto \left( \mathcal{M}_1 \left( x_{\pi(1)} \right), \ldots, \mathcal{M}_n \left( x_{\pi(n)} \right) \right)$$

*is $(\varepsilon_1(\alpha), \delta)$-RDP with*

$$\varepsilon_1(\alpha) = \frac{\log \left( e^{2\alpha^2 \sigma^2} + 4\delta_{\min} e^{\alpha \varepsilon_0} \right)}{\alpha - 1}, \tag{36}$$

*where*

$$\sigma = 8\sqrt{\frac{e^{\varepsilon_0}}{n}} \tag{37}$$

$$\delta_{\min} = e^{-\frac{n}{8(e^{\varepsilon_0}+1)}}. \tag{38}$$

For small $\varepsilon_0$, the result below is useful.

**Lemma I.2** ([40])**.** *Under the same assumptions as Lemma I.1, $\mathcal{S}$ is $(\varepsilon(\alpha), \delta)$-RDP*

$$\varepsilon_1(\alpha) = 2\alpha e^{4\varepsilon_0} (e^{\varepsilon_0} - 1)^2 / n. \tag{39}$$

Applying Lemma G.1, by construction the mapping from each $x_i$ to $y(i, 1), \ldots, y(i, b_0)$ is $(\varepsilon_0, 0)$-LDP. By Lemma I.1, respectively Lemma I.2, the mapping $x^n \mapsto \hat{\mu}^{(k)}$ is $(\varepsilon_1(\alpha), \alpha)$-RDP where $\varepsilon_1(\alpha)$ is given by (36), respectively (39). By composition, Algorithm 4 is $(T\varepsilon(\alpha), \alpha)$-RDP.

We can convert this bound back to $(\varepsilon, \delta)$-DP using Proposition 12 from [30].

**Proposition I.3.** *For all $\delta > 0$, Algorithm 4 is $(\varepsilon, \delta)$-DP where*

$$\varepsilon = \inf_{\alpha \in (1, \infty)} T\varepsilon_1(\alpha) + \frac{\log(1/\delta) + (\alpha - 1)\log(1 - 1/\alpha) - \log(\alpha)}{\alpha - 1}, \tag{40}$$

*where*

$$\varepsilon_1(\alpha) = \min \left( 2\alpha e^{4\varepsilon_0} (e^{\varepsilon_0} - 1)^2 / n, \frac{\log \left( e^{2\alpha^2 \sigma^2} + 4\delta_{\min} e^{\alpha \varepsilon_0} \right)}{\alpha - 1} \right) \tag{41}$$

*and $\sigma, \delta_{\min}$ are given by (37), (38) respectively.*

## J Additional Experiments

Here experiments are done with the same setup as in Section 6, with local vectors $X_i(j) \overset{\text{i.i.d.}}{\sim} \frac{1}{\sqrt{d}} (2 \cdot \text{Ber}(0.8) - 1)$. We set $\delta = 10^{-6}$.

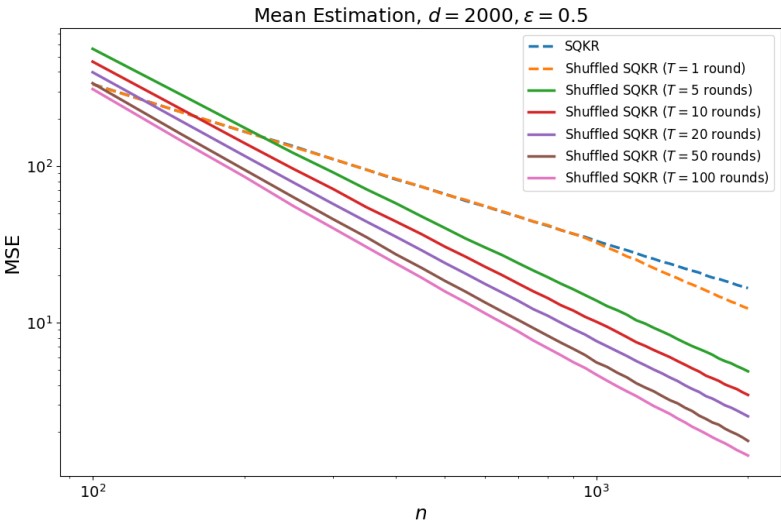

Figure 2: Comparison of MSE vs. number of clients $n$ for LDP scheme (SQKR) and shuffled SQKR. For shuffled SQKR, we set $b_0 = 1$ and choose $\varepsilon_0$ using results in Section I. Communication cost is $\lceil (\log_2(2000) + 1) \rceil = 12$ bits per round.

Figure 2 illustrates separation between Algorithm 4 and LDP schemes. Algorithm 4 achieves error decreasing quadratically with $n$ as guaranteed by Theorem 5.3. With only one round of shuffling, there is separation from the LDP scheme only when $n$ is sufficiently large, and thus order-optimal error performance only occurs for large $n$ (or equivalently small $\varepsilon$). This problem is avoided with multiple rounds of shuffling.

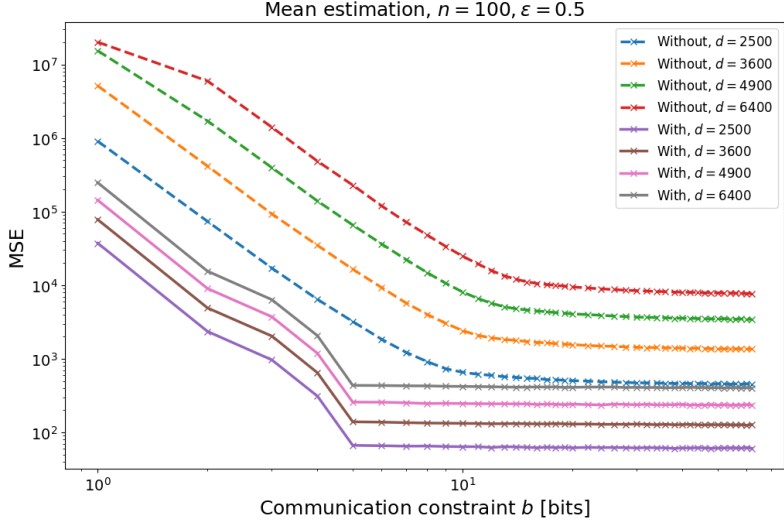

Figure 3: CSGM with and without coordinate pre-selection using $d' = 833$.

Figure 3 compares the performance of CSGM with and without coordinate pre-selection. In this regime coordinate pre-selection improves performance for all $b$. As predicted by Corollary 4.3 and Corollary 4.5, the MSE decreases with $b$ but is effectively constant for sufficiently high $b$ where the

privacy term dominates. We can determine the communication cost needed for order-optimal central DP error performance to be the $b$ at which the MSE is within some fixed constant factor away from the limiting value. We see that the communication cost increases with dimension $d$ with the vanilla CSGM scheme, but a dimension-free communication cost is achieved with coordinate pre-selection.

