# OpenReview forum: "Privacy Amplification via Compression: Achieving the Optimal Privacy-Accuracy-Communication Trade-off in Distributed Mean Estimation"
_NeurIPS.cc/2023/Conference — NeurIPS 2023 poster_

### Official Review · Reviewer_mo9D · 2023-07-06

**Soundness:** 4 excellent
**Presentation:** 3 good
**Contribution:** 3 good
**Rating:** 7
**Confidence:** 4

**Summary:**

This paper studies the distributed mean estimation problem with communication constraints and central DP. Communication constraints are met by subsampling a Kashin representation of the target vector, and \eps-DP is achieved by adding Gaussian noise. Privacy in the absence of a trusted server can be achieved using a LDP mechanism along with a secure shuffler.

After discussions with the authors, I have increased my score.

**Strengths:**

The work claims to be the first to study the communication-privacy-accuracy tradeoff for DME problem with central DP. It provides a simple scheme that achieves order-optimal error bounds.

**Weaknesses:**

While this is the first work to study the problem, the solution was well-known. Unlike the LDP constraints, there is no tension between the communication cost and privacy in this case.  Therefore, in my opinion, the novelty of this work is limited. The results are a straightforward amalgamation of well-known techniques that are independently known to be optimal.

**Questions:**

The results are intuitively correct and are technically sound.

**Limitations:**

Yes

---

> ### Author Rebuttal · Authors · 2023-08-09
>
> We sincerely appreciate the reviewer's acknowledgment of the correctness and technical soundness of our results. However, we kindly disagree with the statement that *"Unlike the LDP constraints, there is no tension between the communication cost and privacy in this case."* Indeed, we believe the situation is exactly analogous to the local DP case, and showing this is part of the contribution of our paper. For local DP, one may a priori expect a tension between privacy and communication cost, e.g.,  privatizing the local messages can increase their entropy and hence make them more difficult to compress. However [Chen et al. 2020] and others have shown that these two constraints can be aligned in a way such that the optimal accuracy is only dictated by one of the two constraints and the less stringent constraint can be satisfied for free. Similarly for central DP, one may a priori expect that compression will increase the sensitivity of the mean and therefore require the addition of larger noise at the central server, and hence lower accuracy, for the same privatization level.
>
> The main contribution of our paper is to show that this does not need to be the case when compression is done randomly and privacy amplification due to random compression is taken into account. We believe this idea of using random compression for privacy amplification is novel and does not appear in the prior literature. In addition, we also characterize the optimal three way trade-off for the shuffling model. While previous research has explored communication efficiency under local DP or secure aggregation, our work stands out as the first to comprehensively characterize the communication-privacy-accuracy trade-off for both the central DP and shuffling models.
>
> [Chen et al. 2020] Breaking the communication-privacy-accuracy trilemma

---

> > ### Comment · Reviewer_mo9D · 2023-08-14
> > **devil's advocate, not an adversary**
> >
> > Thanks for the clean perspective.
> >
> > "Similarly for central DP, one may a priori expect that compression will increase the sensitivity of the mean and therefore require the addition of larger noise at the central server, and hence lower accuracy, for the same privatization level."
> >
> > However, without any local privacy constraints, using Chen et al. (essentially the same random compression as in this work), the optimal accuracy is dictated only by the communication constraint. Now at the central server, adding Gaussian noise to the mean of the compressed data is known to be optimal. Therefore, I was questioning the novelty in terms of the techniques and how they deviate from the straightforward application of Chen et al. and folklore Gaussian mechanism.

---

> > > ### Author Response · Authors · 2023-08-14
> > > **Response the follow-up comment**
> > >
> > > We thank the reviewer for offering additional details regarding the question. We would like to clarify that the straightforward application of the compression in [Chen et al] with the Gaussian mechanism is *strictly sub-optimal*, since the compressed (and properly decompressed) mean has a much higher sensitivity. Consequently, the direct implementation of the standard Gaussian mechanism inherently gives rise to a larger amount of MSE.
> > >
> > >
> > > The main contribution of this paper is to show that in order to achieve optimal privacy-utility trade-off when the local data is compressed, one has to take into account the randomness in the local compression and carefully leverage it to reduce the amount of DP noise.

---

> > > > ### Comment · Reviewer_mo9D · 2023-08-14
> > > >
> > > > Thanks for the clarification. I will increase my score.
> > > >
> > > > I was under the impression that Kashin embedding with randomized rounding followed by coordinate subsampling is exactly what is done by Chen et al. also.
> > > >
> > > > Here is an alternate approach (which I initially understood as your result):
> > > >
> > > > Without any LDP constraints, Chen et al. should give you optimal communication vs accuracy tradeoff for mean estimation.
> > > > And you can further achieve central DP by adding Gaussian noise to the achieved optimal mean estimate.
> > > >
> > > > Where does this approach fail or differ from yours?

---

> > > > > ### Author Response · Authors · 2023-08-14
> > > > > **Response the follow-up question**
> > > > >
> > > > >
> > > > > Thanks for the follow-up questions. The procedure outlined is indeed aligned with our CGSM algorithm, but the key question in this procedure is *”how much Gaussian noise do we need to add to ensure $\varepsilon$-DP?*
> > > > >
> > > > >
> > > > > If one were to calibrate the noise scale (i.e., variance) based on the sensitivity of the compressed mean, an additional factor of $\Omega(d/b)$ emerges in comparison to the sensitivity derived from raw, uncompressed samples. This consequently introduces an extra $\Omega(d/b)$ term into the MSE, where $b$ is the compressed dimension (or communication budget), and $d$ is the original dimension.
> > > > >
> > > > >
> > > > > Our work shows that one can get rid of this $\Omega(d/b)$ factor while still achieving the same level of DP (even though the sensitivity is higher!) by leveraging the randomness introduced during the compression/sampling step . Note that the standard Gaussian mechanism with the noise calibrated to the sensitivity (of the compressed sample) would fail to achieve the optimal privacy-accuracy trade-off. The key idea that our paper brings is that the randomness in the local compression can be leveraged for privacy amplification in central DP (and hence the title of the paper).
> > > > >
> > > > >
> > > > > Lastly, as a side note, our compression scheme coincides with SQKR from [Chen et. al] only when the communication budget $b$ is large enough $b = \Omega(\sqrt{d})$. For scenarios where communication budgets are more limited ($b = O(\sqrt{d})$), a more sophisticated sampling strategy, a more sophisticated sampling strategy is needed (i.e., the coordinate pre-selection scheme presented in Algorithm 2).

---

> > > > > > ### Comment · Reviewer_mo9D · 2023-08-20
> > > > > >
> > > > > > Thank you for the clarification.

---

### Official Review · Reviewer_WRY9 · 2023-07-07

**Soundness:** 3 good
**Presentation:** 3 good
**Contribution:** 3 good
**Rating:** 6
**Confidence:** 3

**Summary:**

This paper studied the optimal rates of the tradeoff among privacy (central DP), utility ($\ell_2$ estimation error), and communication cost (in bits) for the distributed mean and frequency estimation settings. The authors used Kashin's representation and Coordinate Subsampled Gaussian Mechanism to achieve the optimal rates. They also propose a many-round procedure with a shuffling mechanism to achieve the optimal rate if there is a secure shuffler instead of a secure central server. Simulation results are provided to supper the theoretical results of the distributed mean estimation setting.

**Strengths:**

1. This paper provides the optimal rates for the distributed mean estimation given specific privacy and communication constraints, which is a fundamental problem in distributed learning (with a central server).
2. The authors proposed the Coordinate Subsampled Gaussian Mechanism and used it to realize the optimal rates.
3. The theoretical results are supported by the simulation results.

**Weaknesses:**

1. In Theorem 5.3, the rate is $O(C^2 d \ldots)$ while in the appendix Lemma G.1 and Proposition G.2, the rates are $O(c^2 d)$. As Theorem 5.3 is also using Kashin's representation, I think it should be $c=O(\frac{C}{\sqrt{d}})$ as in Line 211. Therefore, there could be something incorrect in the result of Theorem 5.3. However, removing $d$ from the current result also seems to be strange to me.
2. The authors derived the theoretical results for the histogram estimation while not showing corresponding simulation results. They claimed these results in the abstract and introduction as contributions while only showing them in the appendix, which may not be very appropriate.
3. This paper uses the name 'federated learning' many times while the problem being solved is more restricted to the distributed learning  (with a central server).


Minor weaknesses:
1. Citation issues: [33] is cited as 'in submission' while it is already published in AISTATS 2023. [59] and [60] are duplicates.
2. Line 91. $10^6-10^9$ -> $10^6$ to $10^9$.
3. In Theorem 4.1, $\sigma$ should use $\Omega$ notation since we need it large enough to guarantee privacy. Similar misuses are in other results. This can be seen in the transition from $\sigma_1^2$ to $\sigma^2$ in Line 630 where the former used $\Omega$ while the latter used $O$.
4. Line 358. $\lceil \log d\rceil + 1)T$ should be $(\lceil \log d\rceil + 1)T$
5. Line 361. The period is missing in the end of this line.
6. The citation of Theorem B.3 is missing.
7. Line 628. $e_2^\epsilon$ should be $e^{\epsilon_2}$.
8. Line 642. The second term in this equality is random which must be incorrect. Nevertheless, the final result is unaffected.

---
I have read the rebuttal which addressed my questions and the weakness concerns.

**Questions:**

1. Line 212. What does the dot product in $x_i = K \cdot \tilde x_i$ mean? Is it a matrix multiplied by a vector?
2. Are the notations of $(\alpha)$ and $(\beta)$ in Equation (4) in Line 217 misplaced? I think $(\alpha)$ is only the first part of Equation (4) and $(\beta)$ is the remaining two parts. Nevertheless, this seems not to affect the analysis in Section 4.1.
3. Line 354. For the statement of $d=5000$, is it from the previous claim based on the theoretical results '$b$ is independent of $d$'? Or is the result of $d=5000$ missing in this manuscript?
4. Line 622 in Section B. Should the sensitivity of $S_j(x^n)$ be $2c$? Should $Z_{i,j}$ be i.i.d. from $\mathrm{Bern}(\gamma)$ instead of $\mathrm{Bern}(1/\gamma)$?
5. Line 630. Should $\log(d/\delta)$ be $\log(d\gamma/\delta)$? Is $\gamma$ missing here (and elsewhere) intentionally?

**Limitations:**

N.A.

---

> ### Author Rebuttal · Authors · 2023-08-09
>
> We appreciate the reviewer's careful reading, valuable questions, and constructive feedback.
>
> **Response to Weaknesses.**
>
> - We apologize for the inconsistency in notation; the rates in Lemma G.1 and Proposition G.2 should be $O(C^2 d ...)$ instead of $O(c^2 d ...)$, and our main results (Theorem 5.3) are indeed correct as stated. We want to clarify that the results stated in Lemma G.1 and Proposition G.2 are already for the $\ell_2$ geometry, not the $\ell_\infty$ one.
>
> - We chose to move our results on histogram estimation (for federated analytics) to the appendix due to the strict page limit, but if this paper is accepted and we are granted an additional page for the full version, we will definitely include the result for histogram estimation together with simmulation results in the main body of the paper.
>       It is important to note that our scheme is unbiased, so the estimation error is not directly comparable with other frequency estimation schemes (which involve thresholding to reduce estimation error and, hence, are typically biased).
>
> - We believe our setting is indeed well-aligned with standard federated learning settings [Mcmahan et al. 2016, Kairouz et al. 2019]. As described in [Kairouz et al. 2019],
> *"Federated learning (FL) is a machine learning setting where many clients (e.g. mobile devices or whole organizations) collaboratively train a model under the orchestration of a central server (e.g. service provider), while keeping the training data decentralized."*
> To this end, our work focuses on optimizing communication efficiency under central DP or shuffle DP, which are two of the most prominent privacy frameworks for federated learning (another one is local DP which we do not consider in this paper). We characterize the fundamental limits and algorithms to achieve (nearly) optimal communication cost while still achieving the same order of estimation error. Our simulations further support the theoretical results.
>
> [Mcmahan et al. 2016] Communication-efficient learning of deep networks from decentralized data
>
> [Kairouz et al. 2019] Advances and open problems in federated learning
>
> - Lastly, we appreciate the reviewers detailed input about typos and minor issues; we will promptly fix them in the revision.
>
>
> **Response to Questions.**
>
> - Yes, here $K\in\mathbb{R}^\{d\times D\}$ is the matrix representing the tight frame, and $K\cdot \tilde{x}\_i$ is a matrix multiplication. Regarding Kashin's representation and its use in transforming an $\ell_2$ problem to an $\ell_\infty$ one, we will include a section in the appendix that elaborates on this topic. See also our response to Reviewer rG5G. for more details on Kashin's representation.
>
> - We agree with the reviewer's point that the middle term $O\left(\frac{C^2d^2\log(1/\delta)}{n^2b^2}\right)$ can also be viewed as part of the impact of privatization, as it comes from the DP noise (as stated in Theorem 4.1). However, it becomes significant only when local vectors are drastically compressed (i.e., when the sampling rate $\gamma$ is sufficiently small). We will further clarify this aspect in the revision to avoid any ambiguity.
>
> - We apologize for the confusion regarding the experimental results for $d=5000$. We acknowledge that the previous version had these results but was removed due to space constraints. We will add it back in the appendix for a more comprehensive presentation.
>
> - Thank you for bringing the typos to our attention. We will fix them in the revision. The sensitivity in Section B should be $2c$ (due to our adoption of a replacement notion of DP instead of a removal version), and $Z_{ij} \sim \text{Bern}(\gamma)$. That being said, the final results remain unaffected.
>
> - In line 630, we opt to use an upper bound on $\sigma_1^2$ in order to simplify notation. Also note that we are mainly interested in the regime where $\gamma = \Omega( 1/\sqrt{d})$ (which will be used to achieve the dimension-free communication cost as in Theorem 4.4), so asymptotically $\log(d\gamma/\delta) \asymp \log(\sqrt{d}/\delta) \asymp \log(d/\delta)$. We will clarify it in the proof of Theorem 4.1.

---

> > ### Comment · Reviewer_WRY9 · 2023-08-15
> > **Thank you for the response!**
> >
> > The authors have addressed most of my questions. In the two papers [Mcmahan et al. 2016, Kairouz et al. 2019] provided, the definition of federated learning includes several key properties: Non-IID, Unbalanced, Massively distributed, and Limited communication; They also said, 'An unbalanced and non-IID (identically and independently distributed) data partitioning across a massive number of unreliable devices with limited communication bandwidth was introduced as the defining set of challenges.' Therefore, I still think this paper only provides insights on   Distributed Mean Estimation or in general distributed learning, but not more general federated learning (missing the discussion of unbalanced and non-IID settings).
> > I have raised my score from 5 to 6.

---

> > > ### Author Response · Authors · 2023-08-15
> > > **Thank you for increasing the score!**
> > >
> > > We thank the reviewer for valuable suggestions and for increasing the score. We agree with the reviewer on the multifaceted nature of FL and that our work centers specifically on its privacy and communication aspects. We also agree with the reviewer's suggestion that exploring more practical scenarios, such as unbalanced data or personalization, is crucial to enhancing the general understanding of FL's real-world implications.
> > >
> > > We however would like to briefly clarify that our work does not assume local data to be generated i.i.d. Our goal is to estimate empirical means (a canonical formulation ofFedAvg or FedSGD), which does not involve any distributional assumption on local data/gradients.
> > >
> > > Once again, we express our gratitude to the reviewer for their valuable feedback and constructive suggestions.

---

> > > > ### Comment · Reviewer_WRY9 · 2023-08-16
> > > > **Thank you for the response!**
> > > >
> > > > I do not have further questions.

---

### Official Review · Reviewer_rG5G · 2023-07-07

**Soundness:** 4 excellent
**Presentation:** 3 good
**Contribution:** 4 excellent
**Rating:** 6
**Confidence:** 3

**Summary:**

The paper proposes a new randomization scheme for differentially private mean estimation. It is based on each user choosing randomly the coordinates to be sent in their messages. Similar scheme has been proposed in reference paper [55] (Hu et al., 2020), however Hu et al. do not sparsify the individual data-element-wise vectors, only mini-batch gradients.

In the proposed scheme, data-element-wise vectors are sparsified randomly and the server averages these sparse vectors and adds normally distributed noise. A classical differential privacy analysis (meaning it uses some classical (eps,delta)-amplification results) is carried out resulting in optimal communication - privacy - accuracy tradeoffs (solving also an open problem in the untrusted server - setting) when these quantities are measured asymptotically.

Algorithms both in trusted server setting and untrusted server (via multimessage shuffling) are given.

The privacy analysis is essentially for the $\ell_\infty$-mean estimation, and the $\ell_2$-mean estimation bounds are obtained using the so called Kashin's representation which turns $\ell_2$-constraints into $\ell_\infty$-constraints in an optimal way.



**Strengths:**

- solves an open problem and given optimal tradeoffs between privacy, accuracy and communication both in trusted and untrusted setting
- Generally well written
- extensive list of references and a comprehensive 'related works' section. I think the paper serves even as a good starting point for diving into this topic.

**Weaknesses:**

- I think some parts are too densely written. The Kashin's representation is not introduced properly I think, those lines 208 to 214 give simply to little information. Like what is that tight frame $K$? I think there is some (still quite short) introduction in reference [34]. I think it would help a lot to have some more material on Kashin's representation, e.g. in the appendix. You use it repeatedly to get the $\ell_2$ mean estimation bounds anyways.

- Related to the above comment, and for further improving writing: in lines 645-646 in the appendix you write "applying the same trick of Kashin's representation"... what are you referring to with 'the same' ? Here also would help expanding / elaborating a bit.

- I think the paper is not entirely explicit about the difficulty of accurate privacy accounting in case of $\ell_2$-mean estimation (see the question comment below in the Questions - section)

- It is not entirely clear how well the method fits to reducing communication in FL training of ML models if one would need to randomly sparsify individual gradients (see comment in the Questions - section)

**Questions:**

-  You mention that the privacy analysis could possibly be improved, and the log factors shaved off. Somehow I get the impression that you could elaborate on this more... Looking at the trusted server setting, I get the impression that the $\ell_\infty$-mean estimation algorithm one could perhaps analyse tightly (with RDP or directly with approximative DP). In a sense I think you do that in the experimental example. However, again with the $\ell_\infty$-constraints, how would you go about involving the Kashin's representations in the analysis? I get the impression that, e.g., in reference [34], the privacy bounds are simply big-O bounds as there are some universal constant involved when transforming between the constraints. Do you think it would be possible to analyse $\ell_2$-mean estimation algorithm using this coordinate subsampling tightly?

- How would this fit to federated learning? I mean wouldn't the sparsity be lost quickly in case each device/user has model updates constructed using several data-elements? Wouldn't sparsification similar to the that of [55] be then required to reduce the communication?

Minor comments:
- Shouldn't "$\ell_2$ error" in Table 1 be actually "$\ell_2^2$ error ?
- In bibliography, please use the Arxiv reference for the reference [33] instead of "in submission"
- The reference paper [55] (which is quite central here) has been published in IJCAI, you only mention the Arxiv reference.

**Limitations:**

Yes, I think the limitations with the analysis are adequately discussed in Section 7.

---

> ### Author Rebuttal · Authors · 2023-08-09
>
> We appreciate the reviewer's comprehensive summary of our work. We would like to clarify the differences between our techniques and those presented in [Hu et al.]. One of the main distinctions is that Hu et al. consider a mini-batch setting, where sparsification, a form of dimensionality reduction, is performed *after* locally perturbing the mini-batch gradients. Although the $\ell_2$ norm of the noise might appear reduced due to the projection step, constructing an unbiased estimator requires multiplying the sampled gradient by a factor of $1/p$, where $p$ is the sampling rate. Consequently, the $\ell_2$ norm of the noise in the unbiased estimator indeed increases by a factor of $1/\sqrt{p}$, resulting in a suboptimal privacy-communication-accuracy trade-off.
>
> In contrast, our techniques independently apply the sampling step to different clients, and the noise is added to the mean of the decompressed gradients, rather than the original (local) uncompressed gradients. Most significantly, the randomness of local sampling is leveraged in our privacy analysis, enabling a tighter trade-off. As the title of our paper suggests we believe the key observation that our paper brings is that the randomness in the local compression can be leveraged for privacy amplification in central DP. It is important to note that the randomness of the sampling/projection in [Hu et al.] is not taken into account in their analysis.
>
> Additionally, we present a communication-efficient scheme that achieves (nearly) optimal communication cost and MSE under a shuffle DP guarantee.
>
> Response to Weaknesses:
> - Regarding Kashin's representation, roughly speaking, a tight frame can be interpreted as a "basis" (i.e., a collection of $D$ vectors in $\mathbb{R}^d$ with $D > d$) with redundancy, and under this "basis", one can find a representation of any $x \in \mathbb{R}^d$ with small $\ell_\infty$ norm.
>
>     We have kept the overview of Kashin's representation brief in our submission as we felt like this has become a standard tool for DME in the literature, see for example [Chen et al. 2020], but following reviewers' suggestions we will provide a detailed discussion of Kashin's representation and how it can be used to transform the $\ell_2$ problem to an $\ell_\infty$ one in an appendix. Below we provide those details for reviewers' reference.
>
> We first introduce the idea of a tight frame in Kashin's representation. A tight frame is a set of vectors $\\{u_j\\}^D_{j=1} \in \mathbb{R}^d$ that satisfy Parseval's identity, i.e. $ \| x \|^2_2 = \sum_{j=1}^D \left\langle u_j, x \right\rangle^2$ for all $x \in \mathbb{R}^d.$
>
> A frame can be viewed as a generalization of the notion of an orthogonal basis in $\mathbb{R}^d$ for $D>d$.
> To increase robustness, we wish the information to be spread evenly across different coefficients, which motivates the following definition of a Kashin's representation:
>
> **Definition.**
>     For a set of vectors $\\{u_j\\}^D_{j=1}$, we say the expansion  $ x = \sum_{j=1}^D a_j u_j $,  with $\max_j | a_j | \leq \frac{K}{\sqrt{D}}\| x \|_2 $ is a Kashin's representation of the vector $x$ at level $K$ .
>
> [Chen et al. 2020] Breaking the communication-privacy-accuracy trilemma
>
> - Here, we refer to the same technique in the proof of Corollary 4.3 (which extends the $\ell_\infty$ mean estimation problem to $\ell_2$ mean estimation). Again, this is based on Kashin's representation and the details stated in the beginning of our general response can be easily provided in an appendix. We will  also elaborate more in the proof to avoid any confusion.
>
> - For the remaining two points, see our responses to the questions below.
>
> Response to Questions:
>
> - Regarding the theoretical analysis of privacy guarantees, we utilized the strong composition theorem in [Dwork et al. 2010], which is only tight up to some logarithmic factors. We acknowledge that employing more advanced accounting techniques such as the strong composition [Kairouz et al. 2016], moment accountants [Abadi et al. 2016], or Renyi DP accountant [Mironov et al. 2017] may yield slightly better bounds on $\varepsilon$. Indeed, in our experiments, we accounted for the privacy budgets via Renyi DP, which we believe offers a tighter bound than our theoretical (order-wise optimal) bounds.
>
>     Regarding the $\ell_2$ mean estimation in practice, we opted not to use Kashin's representation and instead perform random rotation with $\ell_\infty$ clipping. Although the $\ell_\infty$ clipping step introduces a small amount of bias, we believe it offers a better privacy-MSE trade-off in practice. One may be able to directly analyze the privacy of $\ell_2$ mean estimation  with coordinate-sampling (without the Kashin's representation step), but currently it remains unclear.
>
>     [Dwork et al. 2010] Boosting and differential privacy
>
>     [Kairouz et al. 2016] The composition theorem for differential privacy
>
>      [Abadi et al. 2016] Deep learning with differential privacy
>
>     [Mironov et al. 2017]  Rényi differential privacy
>
> -  Note that in the context of standard FL, it is not guaranteed that each local gradient or model update exhibits sparsity. As a result, additional sparsification procedures must be explicitly introduced to effectively mitigate communication costs. In the work of [55], this sparsification process involves the random selection of $k$ coordinates from a set of $d$ coordinates. Our scheme, the Coordinate Subsampled Gaussian Mechanism, on the other hand, can also be viewed as sparsification (as mentioned earlier). However, our primary contribution lies in demonstrating how this sparsification can be used to "amplify" central DP. In comparison to [Hu et al.], our scheme injects isotropic Gaussian noise after local sparsification and aggregation, which enables us to apply the amplification lemma via subsampling.
>
> Lastly, we appreciate the reviewer's attention to the typos and citation inconsistency, and we will promptly address those issues.

---

> > ### Comment · Reviewer_rG5G · 2023-08-11
> > **Reply**
> >
> > Thanks for your rebuttal. I still would like to ask about the last point. You replied:
> >
> > > Note that in the context of standard FL, it is not guaranteed that each local gradient or model update exhibits sparsity. As a result, additional sparsification procedures must be explicitly introduced to effectively mitigate communication costs. In the work of [55], this sparsification process involves the random selection of  coordinates from a set of  coordinates. Our scheme, the Coordinate Subsampled Gaussian Mechanism, on the other hand, can also be viewed as sparsification (as mentioned earlier). However, our primary contribution lies in demonstrating how this sparsification can be used to "amplify" central DP. In comparison to [Hu et al.], our scheme injects isotropic Gaussian noise after local sparsification and aggregation, which enables us to apply the amplification lemma via subsampling.
> >
> > My point was that wouldn't you need to sparsity every individual data-sample wise gradient if you want to have the privacy amplification from the random sparsification? If a user has several gradients, and sparsities, e.g., the sum, then you would not get the desired amplification for the central guarantee, and on the other hand, if you sparsity individual gradients and sum, you quickly loose the sparsity?

---

> > > ### Author Response · Authors · 2023-08-11
> > > **Reply to Reviewer's follow-up questions**
> > >
> > > Thank you for your prompt response and for providing additional clarification regarding your question. In the context of FL when each user has multiple data samples, the vector $x_i$ in our distributed mean estimation (DME) problem formulation can be taken as the mean local gradient computed at user $i$  (i.e., the average of sample-level gradients computed over $m$ samples at client $i$ for some arbitrary $m$). In other words, our Coordinate Subsampled Gaussian Mechanism and consequently the sampling-and-amplification lemma (Theorem 2) will be applied to the *average* gradient computed at each user and will achieve the order-optimal MSE. Note that the privacy amplification comes from the fact that we randomly sparsify the final vector $x_i$ each user aims to communicate to the server (i.e. a random subset of the coordinates of $x_i$ are communicated to the server). The fact that $x_i$ represents the *average* local gradient at each user does not change the privacy amplification.
> > >
> > > We believe the reviewer's question also relates to the the distinction between *user-level* DP (where two neighboring datasets differ in one *user*, which may hold $m$ samples/gradients) and *item-level* DP (where two neighboring datasets differ in one *sample*). Note that when applied to the average user-level gradient as described above our scheme leads to user-level DP, which is stronger than item-level DP. We provide a more mathematical argument below.
> > >
> > >
> > > Mathematically, under the multi-sample setting, each of the $n$ clients holds $m$ samples and each sample is associated with a (sample-level) gradient $g_{i, j}$, where $i \in [n]$ is the client index and $j \in [m]$ is the sample index. In each round of training, the server aims to estimate $\bar{g} = \frac{1}{nm}\sum_{i=1}^n\sum_{j=1}^m g_{i, j}$.
> > >
> > > Since we focus on *user-level* DP, the $L_2$ sensitivity of $\bar{g}$ is defined as
> > > $$ \max\_{(g\_{i,1},...,g\_{i,m}), (g'\_{i,1},...,g'\_{i,m})} \left\Vert \frac{1}{nm}\sum_{j=1}^m g_{i, j} - \frac{1}{nm}\sum_{j=1}^m g'_{i, j} \right\Vert_2 = \frac{2C}{n}, $$
> > > where we assume each local gradient is clipped to $C$ ahead of time (note that the $\ell_2$ clipping is not needed if the loss function is $C$-Lipschitz).
> > >
> > > If there is no communication constraint, one can apply the standard Gaussian mechanism and achieve a MSE $\mathbb{E}\left[ \left\Vert\bar{g} - \bar{g}_{\text{gauss}} \right\Vert^2_2 \right] = O\left(\frac{C^2d}{n^2\varepsilon^2}\right)$.
> > > On the other hand, to further reduce communication cost, one can also apply our coordinate-sampling algorithm CGSM (Algorithm 2 in our paper) or the shuffled-SQKR scheme (Algorithm 4) to user-level gradients $g\_i := \frac{1}{m}\sum\_{j=1}^m g\_{i, j}$, yielding an $\tilde{O}\left(\frac{C^2d}{n^2\varepsilon^2}\right)$ MSE as well as $\tilde{O}(n\varepsilon^2)$ bits of communication. This means that only user-level subsampling is necessary, and as mentioned by the reviewer, item-level sampling does not help as it does not necessarily sparsify the user-level gradient.
> > >
> > > We note that [55] considers item-level DP (see the derivation of Theorem 1 therein) as opposed to user-level DP as described above. Although their framework could be adapted to user-level DP, such a modification will still fail to leverage the randomness introduced during the sampling step, thus lead to suboptimal communication costs.
> > >
> > >
> > > As a side note, while our current discussion assumes full client participation in every training round with all local samples, our analysis extends to scenarios where clients are initially sampled by the server or where each client computes their local updates on a local mini-batch.  By applying standard privacy amplification by subsampling [Balle et al. 2018], one can obtain an optimal privacy-accuracy trade-off.
> > >
> > >
> > > Finally, we thank the reviewer for bringing this up and we plan to briefly mention the application to FL with multiple-samples per user in the revision as discussed above. We look forward to further discussions and input.
> > >
> > > [Balle et al. 2018] Privacy Amplification by Subsampling: Tight Analyses via Couplings and Divergences

---

> > > > ### Comment · Reviewer_rG5G · 2023-08-16
> > > >
> > > > Thank you, the fact that all results are stated for user-level privacy definitely clarifies things! It's all clear now in that respect.
> > > >
> > > > > Regarding the $\ell_2$ mean estimation in practice, we opted not to use Kashin's representation and instead perform random rotation with $\ell_\infty$ clipping. Although the $\ell_\infty$ clipping step introduces a small amount of bias, we believe it offers a better privacy-MSE trade-off in practice. One may be able to directly analyze the privacy of $\ell_2$ mean estimation with coordinate-sampling (without the Kashin's representation step), but currently it remains unclear.
> > > >
> > > > This is my point. Similarly to the Poisson binomial mechanism, it seems to me that it is unclear how to accurately do the privacy accounting for $\ell_2$ mean estimation. Just like in the Poisson binomial mechanism paper, here the experimental example where RDP is used is such that the $\ell_2$-sensitivity is constant. I would imagine that the Skellam mechanism or the discrete Gaussian mechanism would be competitive (when not considering communication constraints).
> > > >
> > > > Although the Kashin representation might be discussed in the literature, the paper should be a coherent presentation, so please add the required definitions and hopefully some material about the Kashin representation. Also adding something about the random rotation + $\ell_\infty$ clipping would be good too.

---

> > > > > ### Author Response · Authors · 2023-08-20
> > > > > **Reply to the official comment**
> > > > >
> > > > > We thank the reviewer for further suggestions. We will compare with Skellam/discrete Gaussian mechanisms as baselines in the experiment, and will also include materials about Kashin's representation and a discussion regarding the empirical random rotation with an $\ell_\infty$ clipping scheme.

---

### Official Review · Reviewer_B1Jv · 2023-07-10

**Soundness:** 3 good
**Presentation:** 3 good
**Contribution:** 4 excellent
**Rating:** 8
**Confidence:** 3

**Summary:**

This paper studies the federated learning problem within the central differential privacy model, where a trusted central server gathers information from local clients. The primary focus is on minimizing communication costs while ensuring privacy and maintaining accuracy guarantees. To address this challenge, the paper introduces a novel privacy amplification framework. The framework operates by having each local client transmit only a random subsample of its full parameters to the central server, effectively reducing the communication overhead. Simultaneously, the randomness introduced in the subsampling process amplifies the privacy guarantee. Notably, in the specific case of distributed mean estimation, this framework significantly reduces the communication cost from an order of $O(d)$ to $\tilde O(n\min\\{\epsilon, \epsilon^2\\})$, showing a significant improvement over state-of-art result.

**Strengths:**

This paper provides a novel privacy amplification method in central DP setting that optimizes the communication-privacy-accuracy three-way trade-off. Furthermore, the proposed distributed mean estimation also has significant implications. For example, upon setting $x_i$ to be local gradients, this algorithm immediately implies a private distributed SGD algorithm with reduced communication cost.

**Weaknesses:**

The privacy amplification method is restricted to the central DP model and does not apply to LDP model.

**Questions:**

N/A

---

> ### Author Rebuttal · Authors · 2023-08-09
>
> We appreciate the reviewer's recognition of the novelty and significance of our techniques in private SGD and federated learning. As the reviwer notes, our schemes do not amplify local DP since the randomness used for amplification (i.e., random seeds for compression/subsampling) must be known by both the client and the server. We note that this is also true for other  amplification technologies, such as privacy amplification by shuffling or subsampling, which are primarily designed for central DP.

---

### Author Rebuttal · Authors · 2023-08-09

We thank all reviewers for their valuable questions, insightful suggestions, and constructive feedback. We will fix all typos/minor issues in the revision and have provided responses to concerns raised by each reviewer below. We would greatly appreciate if the reviewers could consider updating the scores if the concerns are addressed in our response.

---

### Comment · Area_Chair_WCF6 · 2023-08-20
**clarification about the result**

To authors and reviewers:
From a quick look at the results it seems that the results immediately follow from simple and well-known facts.
First: optimal $\ell_2$ error for $\ell_2$ mean estimation with $(\epsilon,\delta)$-differential privacy in the central and shuffle models is $\frac{\sqrt{d}}/{\epsilon n}$ (ignoring the log factor). This lower bound on the final error implies that one does not really need all the information about the input vectors and can instead replace inputs with outputs of an unbiased estimator. This has been previously used explicitly and implicitly in many private compression protocols. Specifically, if every input vector is replaced with an output of an unbiased estimator with mean squared error $\alpha^2$ this will add  $\alpha/\sqrt{n}$ to the error of the sum. This means that any unbiased estimator with mean squared error of at most $frac{d}{\epsilon^2 n}$ would not change the final error asymptotically. Now there are several known ways to compress the $d$-dimensional input to $\epsilon^2 n$ bits that have mean squared error of  $frac{d}{\epsilon^2 n}$.  One can use the one from Chen et al (Thm 2.1) as part of the SQKR algorithm or random projection with subsamling from this work: https://proceedings.mlr.press/v162/vargaftik22a/vargaftik22a.pdf  This paper uses a variation on these techniques.
The same insight applies to histogram estimation, except the results in this work are already implied via shuffling/aggregation of LDP protocols for the entire range of \epsilon.

Would be great to hear from the authors if they believe that this point captures the essence of their contribution and from reviewers on whether that affects their view of this work.

Beyond theory I'm also not convinced by practical significance of the results. As far as I understand, the setting where $d \gg \epsilon^2 n$ seems atypical. I don't recall seeing it in empirical papers and I imagine that for most problems the resulting error is too large to get practically meaningful estimates.

---

> ### Author Response · Authors · 2023-08-20
> **Response to AC's comment**
>
> We sincerely appreciate the AC for handling and reading our paper and raising the questions. We clarify their concerns below.
>
> We believe the approach outlined by the AC can be summarized as follows: clients first compress individual local vectors $x_i$ into $n\varepsilon^2$ bits, thereby creating a locally unbiased estimator with a Mean Squared Error (MSE) of $\alpha^2 = d/n\varepsilon^2$, and subsequently, the server aggregates these locally compressed vectors. Since the compression error has the same order as the MSE under an $(\varepsilon, \delta)$-DP constraint, the final MSE will not be affected order-wise.
>
> The critical question here is whether the estimate produced by this scheme is still $(\varepsilon, \delta)$-DP. The local compression has increased the sensitivity of the mean by a factor of $d/b$, where $b$ is the communication budget (intuitively, there is less averaging now because each node is communicating less information), and hence the DP noise must increase accordingly to achieve the same $\varepsilon$. If we increase the DP noise accordingly, this leads to an additional $d/b$ in the final MSE for the scheme described by the AC. It is unclear how one can privatize the locally compressed vector without increasing the sensitivity, and hence the MSE compared to the uncompressed case. Indeed, we show that this is *impossible* without carefully leveraging the randomness used in local compression.
>
> Our main result shows that by carefully accounting for the randomness in the local compression, one can significantly reduce the amount of noise needed, *despite the  larger sensitivity of the mean to the local vectors*. Note that this result cannot be obtained from the simple and well-known facts described by the AC, and we showed that obtaining the optimal privacy-communication-accuracy trade-off requires a non-trivial analysis, which is the main contribution of our work.
>
>
> We would also like to point the AC to Reviewer mo9D's follow-up question (who seems to have initially held the same concern as the AC) ( https://openreview.net/forum?id=izNfcaHJk0&noteId=9ar2F1pJtk ) and our response therein ( https://openreview.net/forum?id=izNfcaHJk0&noteId=Y2raU5pGJK ). We hope these discussions clarify our main contributions.
>
> Finally, $d \gg \varepsilon^2 n$ is indeed a practical (and possibly the most relevant) regime in cross-device FL settings. We would like to clarify that since the mean estimation task is invoked in each round of model updating, $n$ here refers to the number of participating clients *per round* (which is typically a small subset of available clients). Similarly, $\varepsilon$ here is the *per-round* privacy budget, which is of the order of $\varepsilon_{final}/\sqrt{T}$ (due to the composition theorem) where $T$ is the total number of training rounds. In practice, the model size $d$ can be on the order of $10^6$ to $10^9$, and the number of per-round clients $n$ ranges from $10^2$ to $10^5$. In addition, if we target a moderate final $\varepsilon$, say $\varepsilon_{final} = 20$, with total training round $T \geq1000$, then the per-round privacy budget $\varepsilon \leq 1$. As a result, $d \ll n\varepsilon^2$ is indeed very relevant and covers most of the cross-device FL regimes.

---

> > ### Comment · Area_Chair_WCF6 · 2023-08-21
> >
> > Thank you for your response. I see that indeed this simpler approach would not work (the sensitivity grows by $\sqrt{d/b}$ not $d/b$ as far as I can tell though). I now appreciate the importance a more subtle algorithm and more interesting analysis that is needed in this case. Thank you for the clarification and sorry about the misunderstanding.

---

> > > ### Author Response · Authors · 2023-08-21
> > > **Thank you!**
> > >
> > > We are pleased to have addressed the AC’s concerns and questions, and we will make sure to reiterate our main contributions in the upcoming version of the paper. The AC is correct that the l2 sensitivity increases by $\sqrt{\frac{d}{b}}$ instead of $\frac{d}{b}$. We thank AC again for their time and efforts in handling and assessing our paper.

---

### Decision · Program_Chairs · 2023-09-21

**Decision:**

Accept (poster)

**Comment:**

This paper proposes an algorithm for distributed high-dimensional mean estimation in L_2 that reduces communication to at most $O(n \min{\epsilon,\epsilon^2\})$. For sufficiently large $d$ this improves on the naive algorithm communicating $d$ bits. The algorithm requires either an appropriate aggregator or just a multi-message shuffler. The analysis relies on privacy amplification achieved via randomly selecting which indices of the input data to send. This is a neat and novel technique that might find additional applications.